# DRIFT: Divergent Response in Filtered Transformations for Robust Adversarial Defense

**Amira Guesmi & Muhammad Shafique**
eBrain Lab, Engineering Department
New York University Abu Dhabi
Abu Dhabi, UAE

## Abstract

Deep neural networks remain highly vulnerable to adversarial examples, and most defenses collapse once gradients can be reliably estimated. We identify *gradient consensus*—the tendency of randomized transformations to yield aligned gradients—as a key driver of adversarial transferability. Attackers exploit this consensus to construct perturbations that remain effective across transformations. We introduce **DRIFT** (Divergent Response in Filtered Transformations), a stochastic ensemble of lightweight, learnable filters trained to actively disrupt gradient consensus. Unlike prior randomized defenses that rely on gradient masking, DRIFT enforces *gradient dissonance* by maximizing divergence in Jacobian- and logit-space responses while preserving natural predictions. Our contributions are three-fold: (i) we formalize gradient consensus and provide a theoretical analysis linking consensus to transferability; (ii) we propose a consensus-divergence training strategy combining prediction consistency, Jacobian separation, logit-space separation, and adversarial robustness; and (iii) we show that DRIFT achieves substantial robustness gains on ImageNet across CNNs and Vision Transformers, outperforming state-of-the-art preprocessing, adversarial training, and diffusion-based defenses under adaptive white-box, transfer-based, and gradient-free attacks. DRIFT delivers these improvements with negligible runtime and memory cost, establishing gradient divergence as a practical and generalizable principle for adversarial defense.

## 1 Introduction

Despite the steady progress of adversarial defenses, most existing strategies collapse under adaptive attacks. Input transformations such as JPEG compression (Guo et al., 2018), randomized ensembles like BaRT (Raff et al., 2019), SIT Guesmi et al. (2021), and even randomized smoothing (Cohen et al., 2019) all share a critical weakness: their defenses still exhibit *gradient consensus*. An adaptive adversary can approximate gradients across these transformations (e.g., via Expectation over Transformation (EoT) (Athalye et al., 2018)) and exploit the consistent directions that emerge, leading to transferable adversarial examples. This vulnerability stems not from insufficient randomness, but from the fact that most stochastic defenses still preserve a coherent, low-variance surrogate gradient landscape.

We argue that true robustness requires not masking gradients, but *destroying their alignment*. If the gradients through different transformations diverge, then an attacker aggregating them obtains noisy and incoherent signals, severely limiting transferability. Crucially, this principle stands in contrast to prior defenses: unlike BaRT, we do not rely on hand-designed random transformations; unlike randomized smoothing, we do not certify robustness by averaging over smooth noise distributions; and unlike obfuscated defenses (Athalye et al., 2018), we remain fully differentiable, avoiding the pitfall of false robustness.

Recent approaches such as DiffPure and DiffDefense (Nie et al., 2022; Silva et al., 2023) attempt to reverse adversarial perturbations using diffusion models, effectively projecting inputs back onto

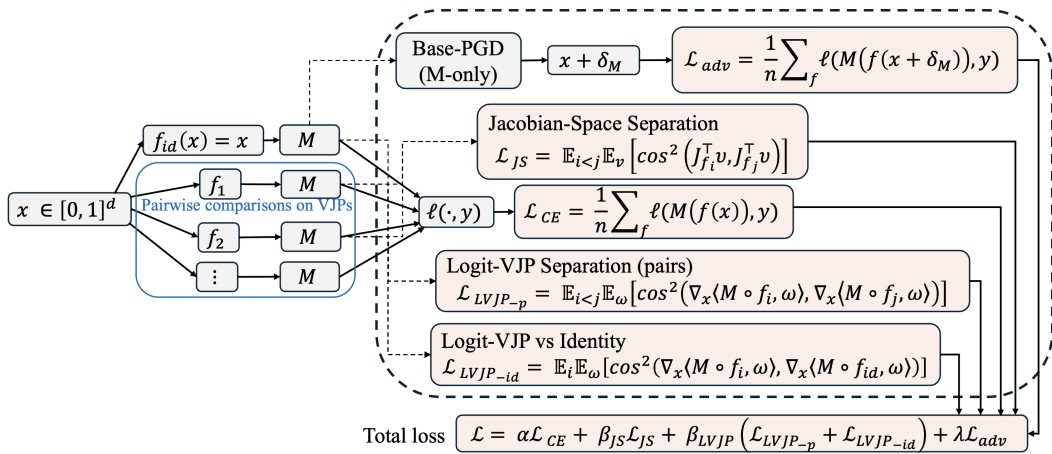

Figure 1: DRIFT methodology. Left: ensemble of learnable filters (plus identity) feeding the frozen base model $M$. Right: separation losses on Jacobian VJPs and logit VJPs (including vs. identity). The base-only PGD and baseline clean performance preservation loss. The total objective sums all terms with weights $(\alpha, \beta_{\text{js}}, \beta_{\text{lvjp}}, \gamma, \lambda)$.

the data manifold. While these methods achieve strong robustness on small- and medium-scale datasets, they are computationally prohibitive for ImageNet-scale tasks and unsuitable for real-time deployment. Moreover, their robustness stems from reconstructing "clean" versions of adversarial examples, which remains vulnerable if the attacker incorporates the purification step into the optimization loop. In contrast, DRIFT is lightweight and online: instead of purifying inputs, we adversarially train differentiable filters that directly disrupt gradient alignment, providing robustness without heavy generative modeling or inference overhead. We introduce **DRIFT** (Divergent Response in Filtered Transformations), a lightweight and architecture-agnostic defense that trains an ensemble of differentiable, learnable filters to explicitly maximize *gradient divergence* while preserving clean accuracy. Each filter is small, efficient, and operates as a front-end to a frozen pretrained model. Through a tailored training objective combining (i) prediction consistency, (ii) Jacobian-space divergence, (iii) logit-space divergence, and (iv) adversarial robustness, DRIFT ensures that while clean predictions remain stable, adversarial optimization encounters conflicting, decorrelated gradient directions. This breaks the attacker's ability to rely on gradient consensus, even under BPDA and EoT.

Our contributions are threefold:

- We formalize the concept of *gradient consensus* and prove that reducing gradient alignment across transformations directly lowers adversarial transferability.
- We propose **DRIFT**, the first differentiable and adversarially trained filter-ensemble defense that explicitly enforces gradient divergence without modifying or retraining the backbone classifier.
- We demonstrate on ImageNet-scale models (ResNet-v2, Inception-v3, DeiT-S, ViT-B/16) that DRIFT consistently outperforms state-of-the-art transformation-based and stochastic defenses against strong adaptive attacks, including PGD-EoT, AutoAttack, transfer-based attacks, and BPDA.

Our findings show that breaking gradient consensus is a general principle for reliable stochastic defenses. By making gradients *diverge* rather than disappear, **DRIFT** provides robustness that is both effective and compatible with real-world, large-scale classifiers.

## 2 RELATED WORK

Defending deep neural networks against adversarial attacks has inspired a broad range of strategies, including input transformations, stochastic preprocessing, adversarial training, architectural modifi-

cations, and generative purification. Below we summarize the most relevant categories, focusing on methods included in our evaluation. Early works rely on fixed transformations to suppress adversarial noise. JPEG compression (Dziugaite et al., 2016) reduces high-frequency perturbations but often degrades clean accuracy and collapses under adaptive white-box attacks. BaRT (Raff et al., 2019) applies randomized blurs, noise, and color shifts at inference to obfuscate gradients. While offering some robustness, these transformations are non-differentiable and not optimized for adversarial resilience, leaving them vulnerable to BPDA-style attacks.

Adversarial Training (AT) (Madry et al., 2018) remains one of the most widely adopted defenses, retraining models directly on adversarial examples. Despite its robustness improvements, AT is computationally costly, tends to reduce clean accuracy, and does not generalize well to unseen threat models. Several variants attempt to mitigate these issues, but the core trade-offs remain. Architectural modifications incorporate robustness into the model design itself. ANF (Suresh et al., 2025) inserts large-kernel convolutional filters and pooling layers at the input to denoise perturbations, but the approach is deterministic and static. Frequency-based strategies such as FFR (Lukasik et al., 2023) regularize filters in the Fourier domain to suppress high-frequency vulnerabilities. These methods improve robustness for CNNs but do not easily transfer to more diverse architectures such as Vision Transformers. Generative purification strategies, such as DiffPure (Nie et al., 2022), reverse adversarial perturbations by projecting inputs back onto the data manifold via score-based diffusion models. While highly effective on small datasets Silva et al. (2023), these approaches are computationally prohibitive for large-scale or real-time scenarios, limiting their practicality.

In contrast to these strategies, our defense **DRIFT** introduces a stochastic, differentiable front-end composed of an ensemble of learnable filters. Crucially, these filters are trained to maximize *gradient divergence* across members, directly disrupting adversarial transferability. Unlike BaRT, DRIFT does not rely on fixed randomness but learns diverse filters optimized for robustness. Unlike adversarial training or diffusion-based purification, DRIFT is lightweight, modular, and plug-and-play with any pretrained classifier, requiring no modification or retraining of the backbone. This makes DRIFT both practical and theoretically grounded against strong adaptive attacks such as BPDA and EOT.

## 3 GRADIENT CONSENSUS ANALYSIS

### 3.1 CORE COMPONENTS

**Notation.** Let $M : \mathbb{R}^d \to \mathbb{R}^K$ be a pretrained classifier that maps an input $x \in \mathbb{R}^d$ to logits $z = M(x) \in \mathbb{R}^K$, with supervised loss $\ell(z, y)$ for label $y \in \{1, \ldots, K\}$. We introduce a bank of lightweight, differentiable, *dimension-preserving* filters $\{f_i\}_{i=1}^n$, each $f_i : \mathbb{R}^d \to \mathbb{R}^d$, and define the filtered pipeline $F_i(x) = M(f_i(x))$. Throughout, $J_g(x) = \frac{\partial g(x)}{\partial x}$ denotes the Jacobian of $g$ at $x$.

**Chain rule and gradient factors.** By the chain rule, the input gradient of the loss through pipeline $F_i$ decomposes as:

$$\nabla_x \ell\big(F_i(x), y\big) = J_{F_i}(x)^\top \nabla_z \ell(z, y) = J_{f_i}(x)^\top J_M\big(f_i(x)\big)^\top \nabla_z \ell(z, y), \qquad (1)$$

with $z = M(f_i(x))$. Eq. 1 makes explicit that adversarial directions are shaped jointly by: (i) the *logit-space* factor $\nabla_z \ell$, and (ii) the *input–output* Jacobians $J_M$ and $J_{f_i}$. DRIFT exploits this factorization by *decoupling* these components across filters, so that shared (consensus) adversarial directions are disrupted.

**Vector–Jacobian products (VJP).** Forming full Jacobians is infeasible in high dimensions. Instead, reverse-mode AD computes $J_g(x)^\top v \in \mathbb{R}^d$, $v \in \mathbb{R}^m$, without ever materializing $J_g(x)$. These VJPs quantify how output directions $v$ backpropagate to the input. Sampling $v$ from Rademacher/Gaussian distributions (Hutchinson probing) yields scalable estimates of Jacobian *(dis)similarity* across filters without storing Jacobians.

**Logit-space probing.** For the composed mapping $x \mapsto z(x) = M(f_i(x)) \in \mathbb{R}^K$, probing with random $w \in \mathbb{R}^K$ gives the logit-space VJP: $g_i(x; w) = \nabla_x \langle z(x), w \rangle = J_{F_i}(x)^\top w$. This provides a tractable surrogate of how each $f_i$ reshapes sensitivity of *decision-space* directions with respect to the input. In practice, $g_i(x; w)$ closely tracks adversarial update directions produced by first-order attacks.

**Adversarial examples and transfer.** An adversarial example $x' = x + \delta$ with $\|\delta\|_p \le \epsilon$ is typically found by iterative first-order methods (e.g., PGD). *Transferability* is the tendency of $\delta$ crafted on a

surrogate to fool a different target. By minimizing cross-filter consensus in Eq. 1, DRIFT makes it harder for a single surrogate-induced direction to generalize across pipelines or models, thus reducing black-box success.

**Dimension-preserving residual filters.** Each filter $f_i$ keeps the input shape and is implemented with a lightweight residual block so $J_{f_i}(x) \in \mathbb{R}^{d \times d}$ is square and $M$ processes $f_i(x)$ without architectural changes. This design keeps runtime small while providing enough flexibility to steer gradient geometry.

## 3.2 GRADIENT CONSENSUS

**Attack Success Probability.** For an input $(x, y)$, let $g_i(x) = \nabla_x \ell(M(f_i(x)), y)$ denote the gradient of the supervised loss through filter $f_i$ and base model $M$. Given a perturbation $\delta$ with $\|\delta\|_\infty \leq \epsilon$, we define the *attack success probability* on pipeline $F_i(x) = M(f_i(x))$ as $p_i(x, \delta) = \mathbb{P}[\arg\max_c M(f_i(x + \delta))_c \neq y]$. A perturbation is transferable if $p_j(x, \delta)$ is high even when $\delta$ was optimized on $f_i$.

**Definition 3.1** (Gradient Consensus). *The* gradient consensus *between two filters $f_i, f_j$ at input $x$ is defined as*

$$\Gamma(f_i, f_j; x) = \left( \frac{\langle g_i(x), g_j(x) \rangle}{\|g_i(x)\|_2 \cdot \|g_j(x)\|_2} \right)^2 .$$

This squared cosine similarity lies in $[0, 1]$. High values indicate that $f_i$ and $f_j$ share adversarially useful directions (high transferability), while low values indicate divergence of gradient subspaces (low transferability).

## 3.3 ASSUMPTIONS

**Assumption 3.2** (Smoothness). *The base model $M$ and filters $\{f_i\}$ are $L$-smooth: their gradients are Lipschitz continuous with constant $L$.*

**Assumption 3.3** (Bounded Gradients). *There exists $G > 0$ such that $\|g_i(x)\|_2 \leq G$ for all $i$ and all inputs $x$.*

## 3.4 THEORETICAL RESULTS

**Lemma 3.4** (Transferability and Consensus). *Let $\delta$ be an adversarial perturbation of size $\|\delta\|_\infty \leq \epsilon$ crafted using gradient $g_i$. Then, under Assumptions 1–2, the expected attack success probability on $f_j$ satisfies $p_j(x, \delta) \leq C \cdot \epsilon G \cdot \Gamma(f_i, f_j; x)$, for some constant $C$ depending on the Lipschitz constant $L$ and the loss margin at $x$.*

*Proof sketch.* A first-order Taylor expansion gives $\ell(M(f_j(x + \delta)), y) \approx \ell(M(f_j(x)), y) + \langle g_j(x), \delta \rangle$. Choosing $\delta$ aligned with $g_i(x)$ yields an inner product $\langle g_j(x), g_i(x) \rangle \|\delta\|_2 / \|g_i(x)\|_2$. By normalizing and squaring the cosine similarity, the transfer effect is scaled by $\Gamma(f_i, f_j; x)$. Boundedness ($\|g_i(x)\| \leq G$) and smoothness ensure the residual terms are controlled, leading to the probability bound up to a constant factor $C$. $\square$

**Theorem 3.5** (Breaking Consensus Reduces Transferability). *Suppose filters $\{f_1, \ldots, f_n\}$ satisfy Assumptions 1–2. If the expected consensus satisfies $\mathbb{E}_{i \neq j}[\Gamma(f_i, f_j; x)] \leq \rho$, $\rho \ll 1$, then for any adversarial perturbation $\delta$ crafted on a single filter $f_i$, its expected transfer success probability on the other filters satisfies $\mathbb{E}_{j \neq i}[p_j(x, \delta)] \leq O(\epsilon G \rho)$.*

**Remark 3.6** (Identity Path). *Including the identity mapping $f_{\text{id}}(x) = x$ in training ensures that adversaries relying solely on $M$'s gradients are also discouraged. This removes the "$M$-only" blind spot and forces robustness even against attacks that ignore filter structure.*

**Connection to DRIFT.** This theory motivates the DRIFT objective: minimizing empirical gradient consensus. Concretely, the Jacobian separation loss $\mathcal{L}_{JS}$ reduces alignment in feature space, and the logit-VJP separation loss $\mathcal{L}_{LVJP}$ reduces alignment in decision space. Together, they enforce the low-$\rho$ regime required by Theorem 3.5, thereby provably reducing transferability of adversarial examples across filters.

## 4 METHODOLOGY: DRIFT

The consensus analysis of Section 3 shows that transferability is controlled by the alignment of gradients across filters. To make adversarial examples non-transferable, we must explicitly reduce this alignment. As shown in Figure 1, DRIFT (Divergent Response in Filtered Transformations) operationalizes this insight by introducing trainable preprocessing filters and losses that enforce Jacobian- and logit-level divergence. Together with adversarial training on base gradients, this creates a system robust to both non-adaptive and adaptive attacks.

### 4.1 LOSS COMPONENTS

DRIFT integrates four complementary objectives designed to balance clean accuracy with robustness and gradient diversity:

**Cross-Entropy Loss.** To preserve baseline predictive performance, we apply standard supervised training across all filters: $\mathcal{L}_{CE} = \frac{1}{K} \sum_{i=1}^{K} \ell(M(f_i(x)), y)$ .

**Jacobian Separation Loss.** To reduce cross-filter transferability, we explicitly penalize alignment between the vector–Jacobian products (VJPs) of different filters:

$\mathcal{L}_{JS} = \mathbb{E}_{i<j} \mathbb{E}_v \big[ \cos^2 \big( J_{f_i}(x)^\top v, \ J_{f_j}(x)^\top v \big) \big]$ , where $v$ is a random probe vector. High $\mathcal{L}_{JS}$ implies shared adversarial directions, which DRIFT suppresses.

**Logit-VJP Separation Loss.** Beyond raw Jacobians, we also enforce divergence at the decision level. By probing the logit space with random directions $w \in \mathbb{R}^K$, we obtain gradients $\nabla_x \langle M(f_i(x)), w \rangle$. We then penalize their pairwise cosine similarity:

$\mathcal{L}_{LVJP} = \mathbb{E}_{i<j} \mathbb{E}_w \big[ \cos^2 \big( \nabla_x \langle M(f_i(x)), w \rangle, \ \nabla_x \langle M(f_j(x)), w \rangle \big) \big]$ . This term ensures that filters remain diverse with respect to how input perturbations propagate into class decisions.

**Adversarial Training Loss.** To maintain robustness under direct attack, we craft adversarial perturbations $\delta_M$ using PGD on the base model $M$ alone. The filters are then trained to resist these perturbations: $\mathcal{L}_{adv} = \max_i \ell(M(f_i(x + \delta_M)), y)$ . This enforces that each filter can withstand attacks crafted in the base model's gradient space.

**Total Objective.** The complete training loss is a weighted combination:

$\mathcal{L} = \alpha \mathcal{L}_{CE} + \beta_{JS} \mathcal{L}_{JS} + \beta_{LVJP} \mathcal{L}_{LVJP} + \lambda \mathcal{L}_{adv}$. By jointly optimizing these terms, DRIFT encourages filters that (i) preserve clean accuracy, (ii) diverge in Jacobian subspaces, and (iii) resist both base-model and cross-filter attacks.

**Filter Architecture.** Each filter $f_i$ is implemented as a lightweight residual convolutional block: a $3 \times 3$ convolution expanding from 3 to 16 channels, a ReLU nonlinearity, and a second $3 \times 3$ convolution projecting back to 3 channels. The filter output is added to the input via a skip connection: $f(x) = x + \mathrm{Conv}_{16 \to 3}(\mathrm{ReLU}(\mathrm{Conv}_{3 \to 16}(x)))$ . This design ensures that each filter remains close to the identity mapping while still being capable of learning meaningful transformations that diversify adversarial gradients. A detailed exploration of the filter architecture (Appendix A.4) and the effect of ensemble size on robustness (Appendix A.5) further support the design decisions of DRIFT.

## 5 EXPERIMENTAL SETUP

**Dataset.** We conduct experiments on the ImageNet dataset (Krizhevsky et al., 2017). Following common practice, we use a randomly selected subset of the validation set for training the filter ensemble, while reserving the remaining portion exclusively for evaluation. This ensures that the filters are trained without access to test samples, thereby providing a fair assessment of robustness.

**Models.** We evaluate DRIFT across both convolutional and transformer-based architectures to highlight its generality. Specifically, we use two widely adopted CNN models: Inception-v3 (Inc-v3) and ResNet-v2-50 (Res-v2) (Szegedy et al., 2016; He et al., 2016). For transformer-based architectures, we include ViT-B/16 (Dosovitskiy et al., 2021) as a representative Vision Transformer and DeiT-S (Touvron et al., 2021), a data-efficient variant trained with distillation.

**Baselines.** We benchmark DRIFT against a comprehensive set of strong baseline defenses spanning multiple categories. *Input preprocessing defenses* include deterministic and stochastic transformations such as JPEG compression (Dziugaite et al., 2016) and BaRT (Raff et al., 2019). *Generative defenses* are represented by diffusion-based purification via DiffPure (Nie et al., 2022). *Architecture-level defenses* include adversarial noise filtering (ANF) (Suresh et al., 2025) and frequency-based regularization strategies such as filter frequency regularization (FFR) (Lukasik et al., 2023). Finally, we include the widely adopted *adversarial training* (AT) (Madry et al., 2018) as a canonical robust-

ness baseline.

**Evaluation Metrics.** We evaluate defense performance using the standard *Robust Accuracy* (RA), defined as the proportion of adversarial examples that are successfully classified by the target model. Higher RA ($\uparrow$) indicates stronger defense and baseline (standard) accuracy, performance of the model in a benign setting (i.e., no attack).

**Parameter Settings.** For training the set of filters, we use four filters formed by two convolution layers , we use $\epsilon = 4/255$. The number of PGD iterations is set to $T = 10$, with a step size of $\eta = \epsilon/T = 0.4/255$. $\alpha = 1$, $\beta_{js} = 0.5$, $\beta_{lvjp} = 0.5$, $\lambda = 1$, $js\_num\_probs = 5$, $lvjp\_num\_probs = 5$, $epochs = 100$. For the optimizer, we use AdamW with $lr = 1e - 3$, $weight\_decay = 1e - 4$.

**Attacks.** We evaluate DRIFT under a comprehensive suite of strong white-box and black-box attacks. *Gradient-based attacks* include the canonical $\ell_\infty$ Projected Gradient Descent (PGD) (Madry et al., 2018), momentum-based MI-FGSM (MIM) (Dong et al., 2018), variance-reduced VMI-FGSM (VMI) (Wang & He, 2021), and gradient-smoothing Skip Gradient Method (SGM) (Wu et al., 2020). To benchmark against state-of-the-art evaluation protocols, we also include AutoAttack (AA) (Croce & Hein, 2020b). *Gradient-free attacks* are represented by the Square Attack (Andriushchenko et al., 2020) and the Fast Adaptive Boundary Attack (FAB) (Croce & Hein, 2020a), which probe robustness without relying on gradient information. Finally, to model adaptive adversaries aware of the defense mechanism, we incorporate BPDA (Backward Pass Differentiable Approximation) and EOT (Expectation over Transformation) (Athalye et al., 2018), which are widely recognized for breaking obfuscated or randomized defenses. For completeness, Appendix A.1 presents the full threat model and a detailed implementation of our method, along with training pseudocode.

## 6 RESULTS AND ANALYSIS

### 6.1 NON-ADAPTIVE ATTACKS

We first evaluate DRIFT in a non-adaptive setting: the adversary has full white-box access to the base classifier (architecture and weights) but is unaware of the deployed defense. Table 1 reports robust accuracy across four backbone models (ResNet-v2, Inception-v3, DeiT-S, and ViT-B/16) against eight commonly used attacks at a fixed perturbation budget of $\epsilon = 4/255$ for $\ell_\infty$-based attacks and $\epsilon = 1$ for $\ell_2$ attacks. The results show that DRIFT consistently preserves baseline performance, unlike JPEG compression and BaRT, which significantly degrade accuracy even in the absence of attacks. For example, on ResNet-v2, DRIFT maintains 84.66% clean accuracy compared to only 44.97% with JPEG at $q = 50$. At the same time, DRIFT provides substantially higher robustness across all attacks. Against AutoAttack, DRIFT achieves 74.30% robust accuracy on ResNet-v2, surpassing DiffPure (67.01%) and far exceeding JPEG (14.29% at $q = 75$). These results demonstrate that DRIFT offers a favorable trade-off: it preserves clean accuracy while achieving state-of-the-art robustness across convolutional and transformer-based models. In contrast, existing preprocessing-based methods either distort the input distribution or over-regularize the model, leading to severe drops in standard performance.

### 6.2 ADAPTIVE ATTACKS

Prior work has shown that many input-transformation defenses collapse under adaptive threat models. In particular, (Athalye et al., 2018) demonstrated that defenses relying on gradient masking or non-differentiability can be bypassed by BPDA (Backward Pass Differentiable Approximation), which substitutes the gradient of the transformation with the identity or Average Pool during backpropagation. Moreover, randomness-based defenses can be defeated with Expectation over Transformation (EOT), where the attacker averages gradients across multiple stochastic passes to approximate the true gradient. We evaluate DRIFT against such adaptive attacks by considering both BPDA and EOT settings. As shown in Table 2, classical input transformations like JPEG and BaRT collapse under adaptive PGD and AutoAttack, with robust accuracy dropping close to zero across all models. Adversarial training variants (AT, FFR+AT, ANF+AT) exhibit partial robustness but incur significant drops in clean accuracy and remain vulnerable when the attacker leverages EOT. DiffPure achieves moderate robustness but is highly sensitive to configuration. For DiffPure, computing full gradients under BPDA+EOT was infeasible on our hardware due to the prohibitive memory required

Table 1: Robust accuracy (%) of various defenses against seven attacks at noise budget $\epsilon = 4/255$ for $\ell_\infty$ and $\epsilon = 1$ for $\ell_2$ attacks. Best results per row block (model) are in **bold**.

| Model | Defense | Config. | No Attack | PGD $\ell_\infty$ | PGD $\ell_2$ | MIM | VMI | SGM | AA | FAB | Square |
|---|---|---|---|---|---|---|---|---|---|---|---|
| ResNet-v2 | JPEG | $q = 75$ | 62.96 | 35.45 | 41.80 | 6.88 | 19.05 | 37.04 | 14.29 | 74.07 | 15.34 |
| | JPEG | $q = 50$ | 44.97 | 41.27 | 62.43 | 23.81 | 29.63 | 40.74 | 8.99 | 65.61 | 8.47 |
| | BaRT | $k = 5$ | 50.79 | 23.28 | 37.57 | 13.76 | 24.34 | 21.87 | 12.70 | 43.39 | 15.34 |
| | BaRT | $k = 10$ | 38.10 | 24.34 | 31.22 | 18.52 | 19.05 | 22.22 | 9.52 | 43.39 | 12.17 |
| | DiffPure | $t^\star = 0.15$ | 67.79 | 65.43 | 70.64 | 48.73 | 45.66 | 47.20 | 67.01 | 63.12 | 62.88 |
| | **Ours** | $n = 4$ | **84.66** | **76.19** | **79.53** | **67.20** | **53.44** | **71.43** | **74.30** | **81.16** | **80.95** |
| Inception-v3 | JPEG | $q = 75$ | 78.31 | 6.88 | 28.77 | 2.65 | 3.17 | 5.82 | 1.23 | 58.90 | 6.30 |
| | JPEG | $q = 50$ | 76.72 | 37.57 | 39.12 | 14.29 | 11.11 | 36.51 | 7.64 | 54.55 | 5.65 |
| | BaRT | $k = 5$ | 61.38 | 27.51 | 29.31 | 20.11 | 26.46 | 8.77 | 34.32 | 9.12 |
| | BaRT | $k = 10$ | 50.79 | 29.63 | 31.20 | 23.28 | 22.21 | 24.87 | 6.45 | 35.65 | 8.44 |
| | DiffPure | $t^\star = 0.15$ | NA | NA | NA | NA | NA | NA | NA | NA | NA |
| | **Ours** | $n = 4$ | **80.96** | **76.83** | **78.74** | **73.50** | **65.00** | **77.83** | **76.50** | **80.10** | **79.89** |
| DeiT-S | JPEG | $q = 75$ | 82.54 | 2.65 | 22.75 | 1.59 | 2.65 | 2.65 | 4.23 | 81.48 | 67.72 |
| | JPEG | $q = 50$ | 80.95 | 19.05 | 51.85 | 8.99 | 10.05 | 17.99 | 28.57 | 81.48 | 66.14 |
| | BaRT | $k = 5$ | 74.07 | 28.57 | 34.92 | 19.05 | 19.58 | 26.98 | 29.57 | 49.74 | 53.97 |
| | BaRT | $k = 10$ | 63.49 | 35.45 | 37.57 | 26.46 | 29.10 | 34.39 | 31.22 | 52.91 | 50.26 |
| | DiffPure | $t^\star = 0.15$ | 73.63 | 61.55 | 67.55 | 47.61 | 45.30 | 60.70 | 63.21 | 58.32 | 57.77 |
| | **Ours** | $n = 4$ | **82.42** | **76.67** | **78.89** | **69.37** | **62.49** | **71.48** | **76.24** | **81.23** | **80.07** |
| ViT-B/16 | JPEG | $q = 75$ | 77.25 | 8.47 | 31.75 | 6.35 | 25.93 | 8.47 | 10.58 | 71.96 | 59.79 |
| | JPEG | $q = 50$ | 74.07 | 35.45 | 57.67 | 21.69 | 35.45 | 35.98 | 38.10 | 69.31 | 58.73 |
| | BaRT | $k = 5$ | 68.78 | 31.22 | 38.10 | 25.40 | 36.51 | 34.39 | 26.98 | 50.79 | 50.26 |
| | BaRT | $k = 10$ | 54.50 | 33.33 | 36.51 | 28.04 | 30.69 | 31.10 | 30.16 | 47.62 | 42.33 |
| | DiffPure | $t^\star = 0.15$ | NA | NA | NA | NA | NA | NA | NA | NA | NA |
| | **Ours** | $n = 4$ | **80.48** | **74.66** | **77.01** | **70.95** | **63.90** | **75.19** | **77.30** | **79.83** | **77.30** |

Table 2: Robust accuracy (%) of various defenses against adaptive PGD, and AutoAttack (AA) ($\epsilon = 4/255$, 40 steps) across four models. The best results in each column are highlighted in **bold**.

| Defense | Adaptive | ResNet-v2 | | | Inception-v3 | | | DeiT-S | | | ViT-B/16 | | |
|---|---|---|---|---|---|---|---|---|---|---|---|---|---|
| | | Clean | PGD | AA | Clean | PGD | AA | Clean | PGD | AA | Clean | PGD | AA |
| JPEG | BPDA + EOT | 44.97 | 0 | 0 | 76.72 | 0 | 0 | 80.95 | 0 | 0 | 74.07 | 0 | 0 |
| BaRT | BPDA + EOT | 50.79 | 6.0 | 0 | 61.38 | 11.23 | 9.4 | 74.07 | 5.2 | 3.1 | 68.78 | 7.31 | 4.67 |
| AT | EOT | 64.37 | 16.32 | 3.12 | 74.4 | 2.4 | 7.3 | NA | NA | NA | NA | NA | NA |
| FFR+AT | EOT | 56.85 | 20.53 | 13.24 | NA | NA | NA | NA | NA | NA | NA | NA | NA |
| ANF+AT | EOT | 61.67 | 25.12 | 24.63 | NA | NA | NA | NA | NA | NA | NA | NA | NA |
| DiffPure | EOT | 67.79 | 36.43 | 40.93 | NA | NA | NA | 73.63 | 37.55 | 43.18 | NA | NA | NA |
| DiffPure | BPDA + EOT | NA | NA | NA | NA | NA | NA | NA | NA | NA | NA | NA | NA |
| Ours | EOT | 84.66 | 53.78 | 50.12 | 80.96 | 50.40 | 49.66 | 82.42 | 48.15 | 47.97 | 80.48 | 56.74 | 54.90 |
| Ours | BPDA + EOT | **84.66** | **60.19** | **58.73** | **80.96** | **53.68** | **51.11** | **82.42** | **57.22** | **55.43** | **80.48** | **64.17** | **61.23** |

to backpropagate through the diffusion sampler (Nie et al., 2022). In contrast, DRIFT maintains both clean accuracy and strong robustness across convolutional and transformer-based models. For example, on ResNet-v2, DRIFT achieves $60.19\%$ robust accuracy against BPDA+EOT PGD and $58.73\%$ against BPDA+EOT AutoAttack, outperforming all baselines by a large margin. On Inception-v3 and DeiT-S, DRIFT sustains over $50\%$ robust accuracy against both PGD and AutoAttack, while JPEG and BaRT collapse below $10\%$. On ViT-B/16, DRIFT achieves $64.17\%$ and $61.23\%$ robust accuracy under PGD and AutoAttack respectively, again substantially higher than all competitors.

## 6.3 DRIFT vs. Randomized Smoothing Baselines

Randomized smoothing (RS) reports *certified* top-1 accuracy for an $\ell_2$ ball of radius $r$, i.e., attack-agnostic guarantees that the prediction is invariant to any perturbation $\|\delta\|_2 \leq r$ (Cohen et al., 2019; Salman et al., 2019). In contrast, **DRIFT** is an empirical defense evaluated with adaptive white-box attacks ($\ell_2$ PGD-EOT); the numbers below are *empirical robust accuracies* under $\ell_2$ attacks at the same radii $r$. Table 3 juxtaposes the standard RS baselines with DRIFT on ImageNet-1K for *ResNet-50* and *ViT-B/16*. On ResNet-50, DRIFT exceeds SmoothAdv RS by $+9.1$ to $+19.5$ points as $r$ grows from 0.5 to 3.0. On ViT-B/16, DRIFT outperforms CAF from $r=0.5$ to 3.0 with gains between $+0.2$ and $+13.4$ points. We stress that RS values are *certificates*, whereas DRIFT values are *empirical* and should not be interpreted as certified guarantees.

## 6.4 Ablation Studies

To better understand the contribution of each component in DRIFT, we conduct ablation experiments by selectively including or excluding loss terms during training. Table 4 reports robust accuracy under both non-adaptive and adaptive PGD attacks for four different models. We begin with a baseline that combines standard cross-entropy loss $\mathcal{L}_{CE}$ with adversarial training $\mathcal{L}_{adv}$. While this setup provides reasonable non-adaptive robustness (e.g., $75.66\%$ on ResNet-v2), it collapses almost completely under adaptive attacks (below $10\%$ across all models). Introducing Jacobian-Space Separation ($\mathcal{L}_{JS}$) substantially improves adaptive robustness, boosting performance to $39.80\%$ on ResNet-v2 and similar gains across other architectures. Logit-VJP Separation ($\mathcal{L}_{LVJP}$) proves even more effective, further elevating

Table 3: Robust Accuracy (%) at $\ell_2$ radius $r$ on ImageNet-1K. ResNet-50 uses SmoothAdv randomized smoothing (Salman et al., 2019; Cohen et al., 2019); ViT-B/16 uses Certifying Adapters (CAF) (Deng et al., 2024). *DRIFT* rows are empirical $\ell_2$ robustness at the same radii (not certificates).

| Model | Method | $\ell_2$ radius $r$ | | | | |
| | | 0.5 | 1.0 | 1.5 | 2.0 | 3.0 |
|---|---|---|---|---|---|---|
| ResNet-50 | SmoothAdv RS | 56.00 | 45.00 | 38.00 | 28.00 | 20.00 |
| ResNet-50 | **DRIFT** | **65.13** | **55.34** | **50.78** | **45.66** | **27.45** |
| ViT-B/16 | CAF (RS-style) | 71.80 | 53.60 | 45.80 | 34.20 | 21.20 |
| ViT-B/16 | **DRIFT** | **71.96** | **64.55** | **51.23** | **47.62** | **30.51** |

*Notes.* RS entries (SmoothAdv/CAF) are *certified* accuracies; DRIFT entries are *empirical* accuracies under $\ell_2$ PGD-EOT attacks at the same radii. Certified and empirical numbers should not be compared as if equivalent guarantees.

adaptive robustness to $47.61\%$ on ResNet-v2 and consistently outperforming $\mathcal{L}_{JS}$ across models. Finally, combining all three components ($\mathcal{L}_{CE}$, $\mathcal{L}_{JS}$, $\mathcal{L}_{LVJP}$, and $\mathcal{L}_{adv}$) yields the strongest defense. This full configuration achieves the highest adaptive robustness across all architectures, with ResNet-v2 at $53.78\%$, Inception-v3 at $50.40\%$, DeiT-S at $48.15\%$, and ViT-B/16 at $56.74\%$. Importantly, this robustness comes at no cost to non-adaptive performance, which remains on par with or slightly better than the baselines.

Table 4: Robust accuracy (%) of different loss component configurations against adaptive PGD across four models.

| Loss Components | ResNet-v2 | | Inception-v3 | | DeiT-S | | ViT-B/16 | |
| | Non-adaptive | Adaptive | Non-adaptive | Adaptive | Non-adaptive | Adaptive | Non-adaptive | Adaptive |
|---|---|---|---|---|---|---|---|---|
| $\mathcal{L}_{CE} + \mathcal{L}_{adv}$ | 75.66 | 3.70 | 77.25 | 2.65 | 79.55 | 9.52 | 76.12 | 8.47 |
| $\mathcal{L}_{CE} + \mathcal{L}_{JS} + \mathcal{L}_{adv}$ | 77.21 | 39.80 | 78.64 | 38.54 | 77.90 | 36.71 | 75.34 | 40.12 |
| $\mathcal{L}_{CE} + \mathcal{L}_{LVJP} + \mathcal{L}_{adv}$ | 76.43 | 47.61 | 77.53 | 45.11 | 78.65 | 40.87 | 76.50 | 49.73 |
| $\mathcal{L}_{CE} + \mathcal{L}_{LVJP} + \mathcal{L}_{JS} + \mathcal{L}_{adv}$ | 76.19 | 53.78 | 78.83 | 50.40 | 78.67 | 48.15 | 74.66 | 56.74 |

## 6.5 GRADIENT-NORM SANITY & FINITE-DIFFERENCE CHECK

To rule out gradient obfuscation, we follow the diagnostic in Athalye et al. (2018); Tramer et al. (2020): for each $(x, y)$ we measure $\|\nabla_x \mathcal{L}(x, y)\|_2$ (defense ON) and compare the directional derivative $\mathbf{v}^\top \nabla_x \mathcal{L}$ against a finite-difference slope. We use *BPDA* (identity surrogate), *EOT* over the defense's stochasticity, common randomness (CRN) across paired evaluations, and *centered* differences, $\frac{\mathcal{L}(x+\eta\mathbf{v},y)-\mathcal{L}(x-\eta\mathbf{v},y)}{2\eta}$, with $\eta \in \{10^{-2}, 10^{-3}, 10^{-4}\}$ and 10 random unit $L_2$ directions per sample. Table 5 summarizes the results (medians/means and 5/95 percentiles across the evaluated subset). Gradients are neither vanishing nor exploding (median 1.67, 5–95% 0.44–5.72). The directional mismatch remains in the $10^{-3}$–$10^{-2}$ range with tight tails (p95 $< 4.12 \times 10^{-2}$), indicating informative, non-masked gradients under our defense.

Table 5: D1 diagnostic on ViT-B/16 for ImageNet (subset). Gradients are well-behaved; the directional-derivative mismatch $\Delta_{\mathbf{v}}$ is small across step sizes with tight tails.

| Input-gradient norms ($\|\nabla_x \mathcal{L}\|_2$) | | |
| median | p05 | p95 |
|---|---|---|
| 1.6677 | 0.4392 | 5.7156 |

| Directional mismatch $\Delta_{\mathbf{v}} = \|\mathbf{v}^\top \nabla_x \mathcal{L} - \frac{\mathcal{L}(x+\eta\mathbf{v})-\mathcal{L}(x-\eta\mathbf{v})}{2\eta}\|$ | | | | |
| $\eta$ | median | mean | p05 | p95 |
|---|---|---|---|---|
| $1 \times 10^{-2}$ | 0.00562 | 0.01184 | 0.00205 | 0.02630 |
| $1 \times 10^{-3}$ | 0.00773 | 0.01229 | 0.00106 | 0.02513 |
| $1 \times 10^{-4}$ | 0.00477 | 0.01570 | 0.00133 | 0.04114 |

*Setup.* Identity BPDA surrogate; expectation-over-transforms (EOT) with common randomness; centered finite differences; 10 random unit $L_2$ directions per sample. Lower $\Delta_{\mathbf{v}}$ is better.

## 6.6 LOSS-LANDSCAPE SMOOTHNESS

To assess whether our defense induces masking artifacts, we visualize the loss surface around input $x$ along two random, orthonormal directions $(\mathbf{u}, \mathbf{v})$ in input space, plotting $\mathcal{L}(x + a\mathbf{u} + b\mathbf{v}, y), (a, b) \in [-\tau, \tau]^2$, on a $41 \times 41$ grid with $\tau = 3/255$. For stochastic components, we evaluate the *expected* loss via EOT-128 and use common randomness (CRN) so every grid point shares the same random filter sequence. This follows best-practice diagnostics for ruling out gradient obfuscation (Athalye et al., 2018; Tramer et al., 2020). Figure 2 shows a smooth, near-planar surface with monotone shading (yellow→purple) and a mild anisotropy (slope larger along one axis), with no checkerboard or plateau artifacts. Reading the colorbar, the total loss variation across the square is small ($\Delta L \approx 4 \times 10^{-5}$–$5 \times 10^{-5}$), consistent with informative (non-vanishing) but well-behaved gradients. The landscape around

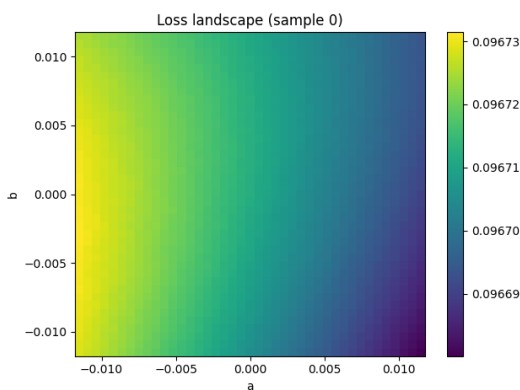

Figure 2: Loss-landscape smoothness. The surface is smooth and nearly planar over $[-3/255, 3/255]^2$ with a coherent slope and no staircase/plateau artifacts.

$x$ is smooth and free of quantization barriers or randomness-induced plateaus, indicating that our defense does not rely on gradient obfuscation per this diagnostic. Further analysis and discussion are provided in appendix A.3.

### 6.7 RUNTIME EFFICIENCY: DRIFT VS. PURIFICATION DEFENSES

Table 6 reports per-image inference cost on ImageNet (ResNet-50). DiffPure Nie et al. (2022) requires 5.52 s, 11.06 s, or 17.07 s per image at timesteps $t^* \in \{0.05, 0.10, 0.15\}$, with $\sim 7.0$ GB GPU memory. In contrast, DRIFT takes only 0.0004 s (0.4 ms) and 0.03 GB. This corresponds to speedups of roughly $1.4 \times 10^4$, $2.8 \times 10^4$, and $4.3 \times 10^4$ over DiffPure at $t^* = 0.05, 0.10, 0.15$, respectively, while using about $\sim 233 \times$ less memory. In sum, DRIFT delivers adaptive robustness with a latency and memory footprint compatible with real-time and resource-constrained settings, whereas diffusion-based purification is orders of magnitude costlier.

Table 6: Inference latency and memory comparison between DiffPure and DRIFT on ImageNet (ResNet-50).

| Defense | Timestep $(t^*)$ | Latency (s) | Memory (GB) |
|---|---|---|---|
| DiffPure | 0.05 | 5.52 | $\sim 7.0$ |
| DiffPure | 0.10 | 11.06 | $\sim 7.0$ |
| DiffPure | 0.15 | 17.07 | $\sim 7.0$ |
| **DRIFT** | **N/A** | **0.0004** | **0.03** |

## 7 CONCLUSION

We introduced **DRIFT**, a lightweight and architecture-agnostic defense framework designed to break *gradient consensus*, a central vulnerability of transformation-based defenses. Unlike prior preprocessing or smoothing methods that preserve coherent gradients and thus remain exploitable under adaptive attacks, DRIFT leverages ensembles of differentiable and learnable filters trained to maximize gradient divergence while preserving clean accuracy. Our theoretical analysis established a formal link between gradient consensus and adversarial transferability, showing that reducing alignment among filters directly limits the success of transferable attacks. Building on this insight, we proposed a principled training strategy that combines cross-entropy, Jacobian separation, logit-VJP separation, and adversarial robustness objectives. Extensive experiments on ImageNet-scale CNN and ViT architectures demonstrated that DRIFT consistently outperforms state-of-the-art preprocessing, adversarial training, and diffusion-based purification defenses. Notably, DRIFT preserved baseline performance under non-adaptive attacks, achieved strong robustness against semi-adaptive and adaptive attacks (including BPDA and EoT), and scaled effectively to both convolutional and transformer backbones. These results underscore DRIFT's practicality as a defense that is efficient, modular, and deployable without retraining or modifying the base classifier.

ACKNOWLEDGMENTS

This work was supported in parts by the NYUAD Center for Cyber Security (CCS), funded by Tamkeen under the NYUAD Research Institute Award G1104.

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

# A  APPENDIX

## A.1  MORE DETAILS OF EXPERIMENTAL SETTINGS

### A.1.1  IMPLEMENTATION DETAILS OF OUR METHOD

**Training Algorithm.** The DRIFT filters are trained jointly with a frozen base model $M$ using a composite loss. Each batch update includes: (i) preserving clean accuracy via cross-entropy; (ii) enforcing gradient diversity through Jacobian separation; (iii) enforcing divergence in logit-space VJPs, including against the identity path; and (iv) adversarial training with base-PGD adversaries. Warmup schedules ensure stability by gradually activating each component. The procedure is summarized in Algorithm 1.

---

**Algorithm 1:** Training DRIFT Filters

**Input:** Frozen base model $M$; trainable filters $\mathcal{F} = \{f_i\}_{i=1}^{K}$; dataloader $\mathcal{D}$; weights
$\quad\quad\alpha, \beta_{\text{JS}}, \beta_{\text{LVJP}}, \gamma$; warmups $w_{\text{js}}, w_{\text{lvjp}}, w_{\text{adv}}$; PGD budget $\epsilon$, steps $T$, step size $\eta$
**Output:** Trained filters $\mathcal{F}$

1 Define identity path $f_{\text{id}}(x) = x$;
2 **for** *epoch* $= 1, \ldots, E$ **do**
3     **for** *batch* $(x, y) \sim \mathcal{D}$ **do**
        // (i) Clean CE across filters
4         $\mathcal{L}_{\text{CE}} \leftarrow \frac{1}{K} \sum_i \text{CE}(M(f_i(x)), y)$;
5         $\mathcal{L} \leftarrow \alpha \cdot \mathcal{L}_{\text{CE}}$;
        // (ii) Jacobian-space separation (after warmup)
6         **if** *epoch* $> w_{\text{js}}$ *and* $K \geq 2$ **then**
7             $\mathcal{L}_{\text{JS}} \leftarrow \text{avg}_{i<j} \, \mathbb{E}_v[\cos^2(J_{f_i}(x)^\top v, \, J_{f_j}(x)^\top v)]$;
8             $\mathcal{L} \leftarrow \mathcal{L} + \beta_{\text{JS}} \cdot \mathcal{L}_{\text{JS}}$;
        // (iii) Logit-VJP separation (after warmup)
9         **if** *epoch* $> w_{\text{lvjp}}$ **then**
10        $\mathcal{L}_{\text{LVJP}} \leftarrow$ pairwise terms + vs-identity terms with random logits $w$;
11        $\mathcal{L} \leftarrow \mathcal{L} + \beta_{\text{LVJP}} \cdot \mathcal{L}_{\text{LVJP}}$;
        // (iv) Base-PGD adversary (after warmup)
12         **if** *epoch* $> w_{\text{adv}}$ **then**
13        $\delta_M \leftarrow \text{PGD}(x, y; M, \epsilon, \eta, T)$;
14        $\mathcal{L}_{\text{adv}} \leftarrow \max_i \text{CE}(M(f_i(x + \delta_M)), y)$;
15        $\mathcal{L} \leftarrow \mathcal{L} + \gamma \cdot \mathcal{L}_{\text{adv}}$;
        // (v) Update
16         Backpropagate $\mathcal{L}$; sanitize NaNs/Infs; clip grads; optimizer step;

---

### A.1.2  IMPLEMENTATION DETAILS OF ADVERSARIAL ATTACKS

**PGD:** We implement $\ell_\infty$ and $\ell_2$ variants of Projected Gradient Descent (PGD) Madry et al. (2018). For $\ell_\infty$ PGD we set $\epsilon = 4/255$, 40 steps with step size $\epsilon/10$. For $\ell_2$ PGD we set $\epsilon = 1$, 40 steps with step size $\epsilon/10$.

**MIM:** The Momentum Iterative Method (MI-FGSM) Dong et al. (2018) introduces a momentum term to stabilize the gradient direction. We use $\epsilon = 4/255$, 40 steps, and momentum decay $= 1.0$.

**SGM:** The Skip Gradient Method (SGM) Wu et al. (2020) applies gradient smoothing to mitigate gradient shattering in residual connections. We set $\epsilon = 4/255$, 40 steps, momentum decay $= 1.0$, and $\lambda = 0.01$.

**VMI:** The Variance-Minimized Iterative Method (VMI-FGSM) Wang & He (2021) reduces gradient variance across steps. We set $\epsilon = 4/255$, 40 steps, momentum decay $= 1.0$, and neighborhood samples $N = 5$.

**AA:** AutoAttack (AA) Croce & Hein (2020b) is an ensemble of parameter-free attacks designed for reliable robustness evaluation. We run the standard configuration with $\epsilon = 4/255$, 40 steps.

**FAB:** The Fast Adaptive Boundary (FAB) Attack Croce & Hein (2020a) minimizes perturbation

norm while crossing the decision boundary. We set $\epsilon = 4/255$ and allow up to $40$ steps.

**Square:** The Square Attack Andriushchenko et al. (2020) is a gradient-free black-box attack using random square-shaped updates. We set $\epsilon = 4/255$ and a query budget of 5000.

**BPDA+EOT:** Following Athalye et al. (2018), we evaluate adaptive white-box attacks using *Backward Pass Differentiable Approximation* (BPDA) combined with *Expectation over Transformation* (EOT). BPDA approximates the backward pass of non-differentiable components with the identity mapping, while EOT averages gradients across multiple stochastic forward passes. We use 40 PGD steps, $\epsilon = 4/255$, and 5 EOT samples per step. This setting provides the attacker full access to the defense while accounting for stochasticity.

### A.1.3 Implementation Details of Adversarial Defenses

**JPEG Compression.** Following Dziugaite et al. (2016), we evaluate JPEG-based defenses with compression quality factors of $75\%$ and $50\%$.

**BaRT.** For BaRT Raff et al. (2019), we apply $k = 5$ and $k = 10$ randomized transformations sequentially at inference time.

**DiffPure.** We use DiffPure Nie et al. (2022) with the authors' recommended hyperparameters for ImageNet-scale evaluation.

**Adversarial Training (AT).** Standard adversarial training (AT) Madry et al. (2018) is included as a baseline for robustness.

**ANF.** For ANF Suresh et al. (2025), we configure the first layer with a convolutional kernel size of $15 \times 15$, 256 filters, and a max-pooling kernel size of $5 \times 5$.

**FFR.** We evaluate Filter Frequency Regularization (FFR) Lukasik et al. (2023) as a frequency-based regularization defense. Since ANF and FFR alone do not yield sufficient robustness on ImageNet, we follow prior work and combine them with adversarial training using a 1-step PGD attack at $\epsilon = 4/255$.

### A.2 More experimental results

**Cross-Filter Transferability.** To assess the transferability of adversarial examples across different trained filters, we employ Projected Gradient Descent (PGD) with $\epsilon = 4/255$ and 40 attack steps. Table 7 shows the robust accuracy (RA) when adversarial samples generated using one filter (source/attacker) are evaluated on another filter (target/victim). The results demonstrate that robust accuracy remains high across all off-diagonal entries, indicating that adversarial perturbations crafted on one filter do not easily transfer to others. This strongly suggests that our Jacobian- and logit-space separation objectives effectively induce gradient divergence among filters, thereby reducing the consensus subspace exploited by transferable attacks. In other words, although each filter alone is capable of mitigating adversarial perturbations, the ensemble diversity ensures that perturbations optimized against one filter fail to generalize to others. This property is critical for our defense strategy, as it directly targets the root cause of adversarial transferability: gradient alignment. The consistently high RA values across ViT-16, ResNet-v2, DeiT-S, and Inception-v3 models confirm the effectiveness of our design in breaking gradient consensus across diverse architectures. Additional analysis and ablation studies are provided in Appendix A.3.

Table 7: Cross-filter transferability reported as robust accuracy (RA %) under PGD ($\epsilon = 4/255$, 40 steps). Rows: source/attacker filter; Columns: target/victim filter.

| Src↓/Tgt→ | ViT-16 | | | | ResNet-v2 | | | | DeiT-S | | | | Inception-v3 | | | |
|---|---|---|---|---|---|---|---|---|---|---|---|---|---|---|---|---|
| | f1 | f2 | f3 | f4 | f1 | f2 | f3 | f4 | f1 | f2 | f3 | f4 | f1 | f2 | f3 | f4 |
| f1 | – | 76.24 | 77.30 | 73.60 | – | 60.32 | 58.73 | 61.38 | – | 61.90 | 61.38 | 56.03 | – | 58.83 | 47.17 | 43.67 |
| f2 | 73.60 | – | 70.95 | 71.00 | 57.67 | – | 59.26 | 59.79 | 58.20 | – | 66.67 | 44.92 | 64.17 | – | 62.67 | 42.50 |
| f3 | 75.10 | 76.24 | – | 72.01 | 58.20 | 59.79 | – | 57.67 | 56.56 | 55.03 | – | 60.85 | 48.83 | 62.83 | – | 61.17 |
| f4 | 73.07 | 76.24 | 74.66 | – | 61.38 | 62.43 | 65.61 | – | 53.92 | 43.92 | 67.20 | – | 45.67 | 46.50 | 60.17 | – |

### A.3 More Analysis

#### A.3.1 Impact of Noise Budget

We assess non-adaptive attacks (the attacker is *not* aware of the defense) using PGD under both $\ell_\infty$ and $\ell_2$ norms. As expected, increasing the perturbation budget degrades accuracy monotonically,

but the drop remains gradual across typical evaluation ranges (See Table 8). These results should be interpreted as an *upper bound* on robustness; see our adaptive evaluations for a stricter assessment.

Table 8: Non-adaptive PGD on ViT-B/16 under $\ell_\infty$ and $\ell_2$ norms.

| | $\ell_\infty$ | | $\ell_2$ |
|---|---|---|---|
| $\epsilon$ | **RA (%)** | $\epsilon$ | **RA (%)** |
| 2/255 | 76.13 | 0.5 | 76.71 |
| 4/255 | 74.66 | 1.0 | 74.07 |
| 8/255 | 68.25 | 1.5 | 73.01 |
| 16/255 | 62.96 | 2.0 | 70.89 |
| 20/255 | 58.62 | 3.0 | 69.30 |

Non-adaptive results provide useful signal about baseline robustness but can overestimate true security; adaptive, defense-aware attacks (e.g., BPDA+EOT) are reported elsewhere in this paper for completeness.

### A.3.2 IMPACT OF EOT

To quantify the effect of expectation over transformation (EOT) on a stochastic defense, we run PGD-$\ell_\infty$ with $K \in \{5, 10, 20\}$ EOT samples per gradient step (common randomness across paired evaluations; all other hyperparameters fixed). As $K$ increases, the attack better estimates the gradient of the *expected* loss and therefore becomes stronger. Table 9 reports Top-1 robust accuracy (%) across $\epsilon$.

(i) Robust accuracy decreases monotonically with $K$ for every $\epsilon$, confirming that EOT makes the attack more faithful to the defense's stochasticity.
(ii) The *marginal* gain from $K=10$ to $K=20$ is smaller than from $K=5$ to $K=10$ at low budgets (e.g., $-1.06$ points at $\epsilon=1/255$), indicating partial saturation; however, at larger budgets the $10 \to 20$ gain remains non-negligible (e.g., $-6.34$ at $\epsilon=8/255$). (iii) Overall drops from EOT-5 to EOT-20 grow with $\epsilon$ (from $-5.82$ at $1/255$ to about $-19$ points at $4/255$–$8/255$), showing that higher budgets benefit more from accurate gradient estimation.

### A.4 IMPACT OF FILTER ARCHITECTURE

The architectural design of the filters $f_i$ crucially influences both their ability to diversify gradients and their efficiency in deployment.

**Expressivity vs. Simplicity.** Filters must be lightweight to avoid prohibitive overhead during inference. At the same time, they need sufficient expressivity to induce distinct gradient directions. For example, a simple residual block with two convolutions and a ReLU nonlinearity

$$f(x) = x + \mathrm{Conv}_2(\mathrm{ReLU}(\mathrm{Conv}_1(x)))$$

already introduces nontrivial nonlinear transformations, ensuring gradients are not trivially aligned. Deeper or wider filters could increase diversity further, but at the risk of overfitting or collapsing to similar functions without additional regularization.

Table 9: **PGD-EOT on ViT-B/16 (ImageNet).** Increasing EOT samples strengthens the attack, lowering robust accuracy. $\Delta$ columns show absolute point drops.

| $\epsilon$ | **EOT-5** | **EOT-10** | **EOT-20** | **$\Delta$ 5→10** | **$\Delta$ 10→20** | **$\Delta$ 5→20** |
|---|---|---|---|---|---|---|
| 1/255 | 81.96 | 77.20 | 76.14 | $-4.76$ | $-1.06$ | $-5.82$ |
| 2/255 | 74.55 | 64.50 | 60.79 | $-10.05$ | $-3.71$ | $-13.76$ |
| 4/255 | 57.62 | 44.39 | 38.57 | $-13.23$ | $-5.82$ | $-19.05$ |
| 8/255 | 37.51 | 24.81 | 18.47 | $-12.70$ | $-6.34$ | $-19.04$ |

**Gradient Geometry.** Architectures with strong local sensitivity (e.g., small convolutional kernels) tend to decorrelate gradients across filters more effectively. Conversely, architectures with large receptive fields or strong averaging effects may inadvertently align filters, weakening separation. Thus, the choice of kernel size, depth, and activation plays a direct role in the effectiveness of $\mathcal{L}_{JS}$ and $\mathcal{L}_{LVJP}$ in promoting divergence.

**Training Stability.** Overly complex filters can destabilize joint optimization: with many parameters, filters may collapse toward learning identity-like transformations, reducing their utility. Lightweight architectures constrain the search space, making separation losses more effective in driving gradient divergence.

**Computation and Deployment.** Inference cost scales linearly with the complexity of each filter. Since DRIFT samples only one filter per forward pass, lightweight architectures preserve the real-time feasibility of the defense. In contrast, heavier filters would diminish the main advantage of DRIFT—efficient deployment without modifying the base model.

**Practical Guidance.** We find that residual-style shallow convolutional filters strike the best balance: they preserve input dimensionality (ensuring model compatibility), introduce sufficient nonlinearity for gradient diversification, and remain efficient at both training and inference. More elaborate architectures (e.g., multi-layer CNNs or attention blocks) may be explored, but lightweight residual filters already achieve strong robustness without sacrificing clean accuracy.

To evaluate how the choice of filter architecture influences DRIFT, we compare several lightweight designs while keeping the base model (ViT-B/16) and training setup fixed. Each filter preserves input dimensionality to maintain compatibility with the backbone.

**Architectures tested.**

- **SingleConv:** A single $3 \times 3$ convolution with ReLU.
- **ResBlock (ours):** Two $3 \times 3$ convolutions with ReLU and a residual skip connection.
- **DeepConv:** A stack of four $3 \times 3$ convolutions with ReLU activations.
- **MLPFilter:** A shallow two-layer MLP applied patchwise to image embeddings.

Table 10: Effect of filter architecture on clean and robust accuracy (%) with ViT-B/16 under adaptive PGD ($\epsilon = 4/255$, 40 steps).

| Filter Architecture | Clean Accuracy | Robust Accuracy |
|---|---|---|
| SingleConv | 77.3 | 42.5 |
| DeepConv | 75.8 | 47.9 |
| MLPFilter | 74.9 | 44.2 |
| ResBlock (ours) | **80.5** | **56.7** |

Results show that the residual block (our chosen design) achieves the best balance: it maintains clean accuracy close to the undefended baseline while significantly improving robust accuracy. SingleConv lacks sufficient expressivity and yields poor robustness. DeepConv provides moderate robustness gains but at the cost of lower clean accuracy and higher training instability. Patchwise MLPs decorrelate gradients somewhat but underperform compared to convolutional filters. Overall, lightweight residual filters offer the most effective and efficient choice for DRIFT.

A.5    IMPACT OF THE NUMBER OF FILTERS

The number of filters $K$ plays a critical role in shaping the defense's effectiveness.

**Gradient Diversity.** Each filter induces a distinct gradient geometry. Increasing $K$ expands the ensemble of gradient subspaces, which makes it harder for an adversary to find perturbations that transfer across all filters. In theory, if the pairwise consensus $\Gamma(f_i, f_j)$ is kept small, the expected transfer success of perturbations decreases roughly as $\mathcal{O}(1/K)$, since adversaries must overfit to one

filter at a time. Thus, larger $K$ strengthens robustness by increasing gradient divergence opportunities.

**Robustness–Accuracy Trade-off.** While more filters can improve adversarial robustness, they may also introduce redundancy or training instability if $K$ is too large. Empirically, small ensembles ($K = 2$–4) already capture strong divergence effects without hurting clean accuracy. Beyond a certain point, gains diminish: additional filters may converge to similar behaviors unless carefully regularized by $LJS$ and $LLVJP$.

**Training and Inference Cost.** The cost of training grows quadratically with $K$ for pairwise losses:

$$\text{Cost} \sim \binom{K}{2}(P_v + P_w).$$

Thus, increasing $K$ both increases backprop passes (for each probe) and the number of pairwise comparisons. Inference, however, remains efficient: only one filter is sampled and applied per forward pass, so runtime cost grows only linearly with $K$ when switching filters randomly. This makes DRIFT scalable in deployment even with moderate $K$.

**Practical Guidance.** We find $K = 3$–5 filters offers the best balance: sufficient gradient diversity to suppress transferability, while keeping training cost manageable. Larger ensembles can be explored if resources allow, but diminishing returns beyond $K = 6$ suggest focusing instead on improving separation losses.

## A.6 PROBE COUNT: VARIANCE–COMPUTE TRADE-OFF

**Estimators.** DRIFT uses Hutchinson-style probing to estimate pairwise alignment. For two filters $(f_i, f_j)$, let

$$\phi_v(x; f_i, f_j) = \cos^2\big(J_{f_i}(x)^\top v, \ J_{f_j}(x)^\top v\big),$$
$$\psi_w(x; f_i, f_j) = \cos^2\big(\nabla_x\langle M(f_i(x)), w\rangle, \ \nabla_x\langle M(f_j(x)), w\rangle\big),$$

with $v$ a unit probe in the filter output space, and $w$ a unit probe in the logit space. With $P_v$ i.i.d. probes $\{v_p\}_{p=1}^{P_v}$ and $P_w$ i.i.d. probes $\{w_q\}_{q=1}^{P_w}$, the Monte Carlo estimators used in $\mathcal{L}_{JS}$ and $\mathcal{L}_{LVJP}$ are

$$\widehat{\Gamma}_{JS}(x; f_i, f_j) = \frac{1}{P_v}\sum_{p=1}^{P_v}\phi_{v_p}(x; f_i, f_j), \qquad \widehat{\Gamma}_{LVJP}(x; f_i, f_j) = \frac{1}{P_w}\sum_{q=1}^{P_w}\psi_{w_q}(x; f_i, f_j).$$

**Concentration.** Each summand is bounded: $\phi_v \in [0, 1]$ and $\psi_w \in [0, 1]$. Hence, by Hoeffding's inequality, for any $\epsilon > 0$,

$$\Pr\Big(|\widehat{\Gamma}_{JS} - \mathbb{E}[\phi_v]| \geq \epsilon\Big) \leq 2\exp\big(-2P_v\epsilon^2\big), \qquad \Pr\Big(|\widehat{\Gamma}_{LVJP} - \mathbb{E}[\psi_w]| \geq \epsilon\Big) \leq 2\exp\big(-2P_w\epsilon^2\big).$$

Consequently, the mean-squared error (MSE) scales as $\mathcal{O}(1/P_v)$ and $\mathcal{O}(1/P_w)$, respectively. In practice, this means doubling the number of probes reduces the estimator's standard deviation by roughly $1/\sqrt{2}$.

**Bias.** The estimators are unbiased for the *probe-averaged* consensus, i.e., $\mathbb{E}[\widehat{\Gamma}_{JS}] = \mathbb{E}[\phi_v]$ and $\mathbb{E}[\widehat{\Gamma}_{LVJP}] = \mathbb{E}[\psi_w]$, where the expectation is over the probe distributions (Rademacher or Gaussian, normalized). Any residual bias relative to *full* Jacobian/gradient alignment comes from using random projections instead of exhaustively scanning all directions; this bias decreases as probe count grows.

**Compute cost.** Each probe requires a VJP/gradient evaluation. Thus the per-batch cost scales linearly in probes:

$$\text{Cost per batch} \approx \mathcal{O}\Big(P_v \cdot \binom{K}{2} + P_w \cdot \binom{K}{2} + P_w \cdot K\Big),$$

corresponding to JS pairs, LVJP pairs, and LVJP-vs-identity terms. Larger $K$ or probe counts improve statistical stability but increase backprop passes.

**Practical guidance.** We find a simple schedule balances stability and throughput:

1. **Warmup (epochs** $1$–$w$**):** $P_v=2$ and $P_w=2$ (fast, lets filters stabilize).
2. **Main training:** $P_v=5$ and $P_w=5$ (good variance–compute trade-off).
3. **High-fidelity refinement (last** $10$–$20\%$ **epochs, optional):** $P_v=8$–$10$, $P_w=8$–$10$ if compute permits.

Empirically, $P_v, P_w \in [5, 10]$ yield stable separation signals and stronger cross-filter non-transferability, while larger values show diminishing returns relative to their linear compute cost.

## A.7 PROOF OF LEMMA 3.4 AND THEOREM 3.5

We provide full derivations for Lemma 3.4 and Theorem 3.5. Recall that for filter $f_j$ we define

$$\ell_j(x) \triangleq \ell(M(f_j(x)), y), \qquad g_j(x) \triangleq \nabla_x \ell_j(x).$$

**Assumptions.** We assume the following standard regularity conditions for all filters $f_j$ and inputs $x$:

- **A1 (Gradient boundedness).** There exists $G > 0$ such that $\|g_j(x)\|_2 \leq G$.
- **A2 ($L$-smoothness).** The loss is $L$-smooth in $x$, i.e.,

$$\|\nabla_x \ell_j(x) - \nabla_x \ell_j(x')\|_2 \leq L\|x - x'\|_2.$$

  Equivalently, for any $\delta$,

$$\ell_j(x + \delta) \leq \ell_j(x) + \langle g_j(x), \delta \rangle + \frac{L}{2}\|\delta\|_2^2.$$

- **A3 (Margin–loss link).** For correctly classified $x$, misclassification requires a loss increase of at least a margin $\kappa(x) > 0$. For inputs under consideration, we assume $\kappa(x) \geq \kappa_{\min} > 0$.

**Gradient consensus.** For filters $f_i$ and $f_j$, define the gradient consensus at $x$ as

$$\Gamma(f_i, f_j; x) = \frac{\langle g_i(x), g_j(x) \rangle}{\|g_i(x)\|_2 \|g_j(x)\|_2} \in [-1, 1].$$

We consider perturbations aligned with filter $f_i$:

$$\delta_i(x) = \varepsilon \frac{g_i(x)}{\|g_i(x)\|_2}, \qquad \|\delta_i(x)\|_2 = \varepsilon.$$

### A.7.1 PROOF OF LEMMA 3.4

**Lemma 3.4.** *Under A1–A2, for any $i \neq j$ and any $\varepsilon \in (0, \varepsilon_{\max}]$,*

$$\ell_j(x + \delta_i(x)) - \ell_j(x) \leq C \varepsilon G \Gamma(f_i, f_j; x)$$

*for some constant $C = 1 + \frac{L\varepsilon_{\max}}{2G}$ depending only on $G$, $L$, and $\varepsilon_{\max}$.*

*Proof.* Using $L$-smoothness (A2),

$$\ell_j(x + \delta_i) \leq \ell_j(x) + \langle g_j(x), \delta_i \rangle + \frac{L}{2}\|\delta_i\|_2^2. \tag{1}$$

**Linear term.** Since $\delta_i = \varepsilon\, g_i/\|g_i\|_2$,

$$\langle g_j(x), \delta_i(x) \rangle = \varepsilon \frac{\langle g_j(x), g_i(x) \rangle}{\|g_i(x)\|_2} = \varepsilon\|g_j(x)\|_2 \Gamma(f_i, f_j; x).$$

Applying $\|g_j(x)\|_2 \le G$ (A1),

$$\langle g_j(x), \delta_i(x) \rangle \le \varepsilon G \Gamma(f_i, f_j; x). \tag{2}$$

**Remainder term.** Because $\|\delta_i\|_2^2 = \varepsilon^2 \le \varepsilon_{\max}\varepsilon$,

$$\frac{L}{2}\|\delta_i\|_2^2 \le \frac{L}{2}\varepsilon_{\max}\varepsilon. \tag{3}$$

Substituting (2) and (3) into (1),

$$\ell_j(x + \delta_i) - \ell_j(x) \le \varepsilon G \Gamma(f_i, f_j; x) + \frac{L}{2}\varepsilon_{\max}\varepsilon.$$

Define

$$C = 1 + \frac{L\varepsilon_{\max}}{2G},$$

giving

$$\ell_j(x + \delta_i) - \ell_j(x) \le C\,\varepsilon G \Gamma(f_i, f_j; x).$$

$\square$

### A.7.2 FROM LOSS INCREASE TO TRANSFERABILITY

Define the misclassification indicator

$$\mathbf{1}_{\mathrm{mis}}^{(j)}(x, \delta_i) = \mathbb{I}\big[M(f_j(x + \delta_i)) \ne y\big].$$

From A3, misclassification requires

$$\ell_j(x + \delta_i) - \ell_j(x) \ge \kappa(x).$$

Using Lemma 3.4,

$$\mathbf{1}_{\mathrm{mis}}^{(j)}(x, \delta_i) \le \mathbb{I}[C\varepsilon G|\Gamma(f_i, f_j; x)| \ge \kappa(x)].$$

Assuming $\kappa(x) \ge \kappa_{\min}$,

$$\mathbf{1}_{\mathrm{mis}}^{(j)}(x, \delta_i) \le \mathbb{I}\Big[|\Gamma(f_i, f_j; x)| \ge \frac{\kappa_{\min}}{C\varepsilon G}\Big].$$

Taking expectation over $x$,

$$p_j(x, \delta_i) \triangleq \mathbb{E}_x[\mathbf{1}_{\mathrm{mis}}^{(j)}(x, \delta_i)] \le C'\varepsilon G\,\mathbb{E}_x[|\Gamma(f_i, f_j; x)|],$$

where $C'$ absorbs $\kappa_{\min}^{-1}$.

### A.7.3 PROOF OF THEOREM 3.5

**Theorem 3.5.** *Let $p_j(x, \delta_i)$ be the transfer probability for filter pair $(i, j)$. Then under A1–A3,*

$$\mathbb{E}_{i \ne j}[p_j(x, \delta_i)] \le C\varepsilon G\,\mathbb{E}_{i \ne j}[|\Gamma(f_i, f_j; x)|].$$

*Thus, reducing the average gradient-consensus directly reduces expected transferability.*

*Proof.* For each fixed $(i, j)$, the bound above gives

$$p_j(x, \delta_i) \le C\varepsilon G\,\mathbb{E}_x[|\Gamma(f_i, f_j; x)|].$$

Taking expectation over the uniform draw of pairs $i \ne j$ and using linearity,

$$\mathbb{E}_{i \ne j}[p_j(x, \delta_i)] \le C\varepsilon G\,\mathbb{E}_{i \ne j}\mathbb{E}_x[|\Gamma(f_i, f_j; x)|] \tag{2}$$

$$= C\varepsilon G\,\mathbb{E}_x\mathbb{E}_{i \ne j}[|\Gamma(f_i, f_j; x)|]. \tag{3}$$

Since $C$ is independent of $(i, j)$ and $x$, this yields

$$\mathbb{E}_{i \ne j}[p_j(x, \delta_i)] \le C\varepsilon G\,\mathbb{E}_{i \ne j}[|\Gamma(f_i, f_j; x)|].$$

$\square$

This completes the proofs of Lemma 3.4 and Theorem 3.5.

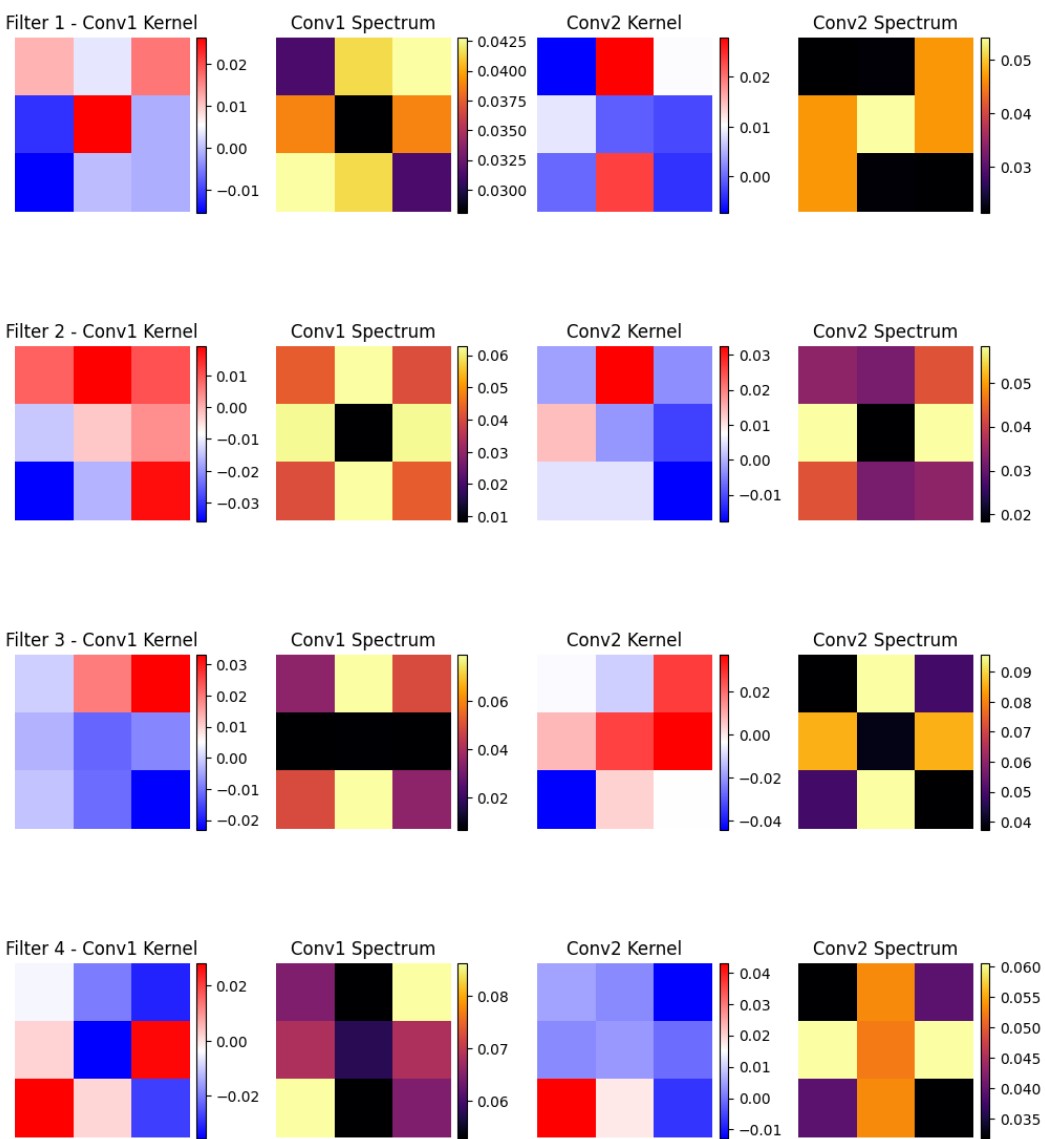

Figure 3: **Learned DRIFT filter kernels (spatial and frequency views).** Each row corresponds to one filter in the ensemble. Left: spatial kernels from the Conv1 and Conv2 layers. Right: their corresponding Fourier magnitude spectra. The filters exhibit complementary spatial–frequency behaviors (e.g., low–mid–frequency smoothing, edge-emphasizing high-frequency responses, and anisotropic patterns), confirming that the ensemble learns diverse transformation regimes rather than collapsing to similar operators.

## A.8 FILTERS INTERPRETABILITY

Figure 3 visualizes the learned DRIFT filters. Each FilterB contains two convolutional layers, whose spatial kernels and frequency spectra exhibit distinct structural patterns. The first-layer kernels (Conv1) emphasize low- and mid-frequency components, while the second-layer kernels (Conv2) introduce sharper, high-frequency structures. This diversity confirms that different filters specialize in complementary spatial–frequency transformations, thereby decorrelating gradient directions across the ensemble.

Figure 4 illustrates the visual transformations produced by each of the four learned DRIFT filters on various ImageNet samples. Despite preserving the overall semantic content, each filter intro-

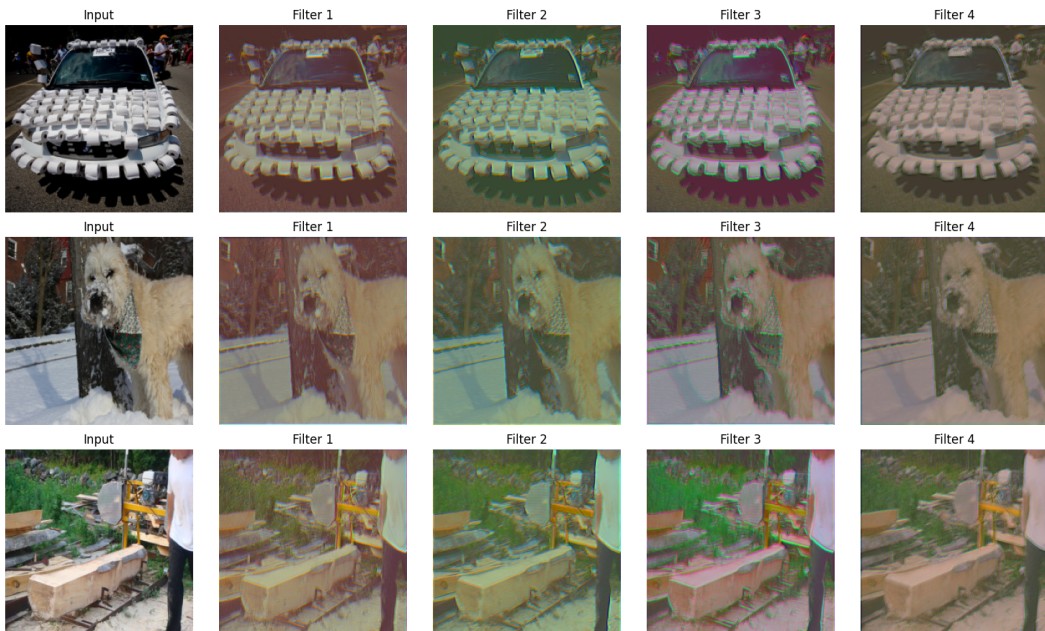

Figure 4: **Qualitative transformations induced by DRIFT filters.** Each column shows the output of a different filter applied to diverse ImageNet samples. Despite preserving semantic content, the filters produce noticeably different low- and high-frequency alterations, including edge enhancement, color–tone shifting, texture smoothing, and contrast modulation. These complementary transformations contribute to DRIFT's ability to decorrelate gradients across filters while maintaining perceptual fidelity.

duces distinct textural and chromatic variations—e.g., Filter 1 enhances local contrast and sharpens edges, producing stronger high-frequency details. Filter 2 modulates mid-frequency color and texture components, introducing characteristic color-tinting and texture amplification. Filter 3 applies mild anisotropic smoothing with a subtle directional bias, reducing fine details in a non-uniform way. Filter 4 performs global brightness and tone lifting while preserving or slightly enhancing local structural details. This diversity of responses confirms that the filters operate in complementary transformation regimes, generating decorrelated input gradients while maintaining perceptual fidelity—an essential factor behind DRIFT's robust, non-obfuscated defense behavior.

