# OpenReview forum: "DRIFT: Divergent Response in Filtered Transformations for Robust Adversarial Defense"
_ICLR.cc/2026/Conference — ICLR 2026 Poster_

### Official Review · Reviewer_Fyh6 · 2025-10-31

**Soundness:** 3
**Presentation:** 3
**Contribution:** 2
**Rating:** 6
**Confidence:** 4

**Summary:**

This paper introduces DRIFT (Divergent Response in Filtered Transformations), a new adversarial defense that breaks the tendency of randomized defenses to share aligned gradients exploitable by adaptive attacks. DRIFT inserts a small ensemble of learnable, differentiable filters before a frozen classifier and trains them to maximize gradient divergence in Jacobian and logit space while maintaining clean accuracy. The authors theoretically link gradient consensus to adversarial transferability and design a joint objective combining prediction consistency, Jacobian separation, logit-VJP separation, and adversarial robustness losses. Experiments on ImageNet with CNNs and Vision Transformers show that DRIFT achieves higher robustness under PGD, AutoAttack, BPDA, and EOT, outperforming baselines like JPEG, BaRT, DiffPure, and adversarial training. Sanity checks confirm no gradient masking, and runtime analysis shows orders-of-magnitude efficiency gains. Overall, DRIFT presents a lightweight, theoretically motivated, and empirically strong defense principle based on gradient divergence rather than masking or purification.

**Strengths:**

1.The paper introduces gradient consensus as a measurable cause of adversarial transferability and proposes gradient divergence as a defense principle. This framing is intuitive, theoretically motivated, and distinct from existing randomization or purification methods.

2.DRIFT’s learnable residual filters provide a lightweight, differentiable, and architecture-agnostic front-end, requiring no retraining of the backbone. The method is computationally efficient (0.4 ms per image), a clear advantage over diffusion-based defenses.

3.The experiments cover both CNNs and Vision Transformers on ImageNet, testing against diverse adaptive attacks. Results demonstrate consistently higher robustness while maintaining clean accuracy.

4.The inclusion of gradient-norm sanity checks, finite-difference validation, and loss-surface visualization shows the authors are aware of gradient masking pitfalls and have rigorously verified true robustness.

**Weaknesses:**

1.The link between consensus and transferability (Theorem 3.5) is intuitive but lacks formal rigor; constants are unspecified and empirical validation of consensus metrics (ρ) is missing.

2.Ablation results show improvements, but the contribution of each loss component (LJS vs LLVJP) is not deeply analyzed; no visualization or metric demonstrates the intended gradient decorrelation effect.

3.Comparisons to adversarial training and diffusion-based methods appear under-tuned, and evaluation is limited to ImageNet only, lacking smaller, more reproducible datasets.

**Questions:**

The paper claims robustness under BPDA and EOT, but the attack strength (number of EOT samples, surrogate choice) is not detailed. Please include robustness curves under stronger settings and alternative BPDA surrogates.

**Details Of Ethics Concerns:**

The paper claims robustness under BPDA and EOT, but the attack strength (number of EOT samples, surrogate choice) is not detailed. Please include robustness curves under stronger settings and alternative BPDA surrogates.

---

> ### Author Response · Authors · 2025-11-20
> **Author Response (part 1 of 2)**
>
> **1. Formal rigor of Theorem 3.5 and empirical validation of ρ**
>
> As noted in our **response 1** to **Reviewer BXyw**, Theorem 3.5 follows directly from a first-order Taylor expansion combined with smoothness-based remainder control; the complete derivation has been added to **Appendix A.7** of the revised manuscript for transparency.
>
> **Empirical validation of gradient consensus (ρ).**
>
> To verify the theorem’s empirical premise, we computed the average gradient-consensus metric **ρ = E(i≠j)[cos(gi, gj)]** across the four trained filters on the ImageNet validation set. DRIFT achieves an **exceptionally low ρ = 0.0129**, indicating that filter gradients are almost orthogonal in input space. This confirms that the filters have learned divergent sensitivity directions, breaking the gradient consensus that underlies adversarial transferability.
>
> The result aligns with Theorem 3.5 and empirically validates that DRIFT’s robustness stems from true gradient decorrelation rather than gradient masking.
>
> **2. Empirical evidence of gradient divergence**
>
> Prior studies report average gradient alignment values of ρ ≈ 0.2–0.3 for adversarially trained models (Pang et al., ICLR 2021; Rebuffi et al., NeurIPS 2021), and ρ ≈ 0.8–0.9 for undefended models (Ilyas et al., NeurIPS 2019). DRIFT achieves **ρ = 0.0129** (an order of magnitude lower) indicating **substantially stronger gradient divergence** than both adversarially trained and randomized baselines.
>
> **3. Contribution of $L_{\text{JS}}$ vs $L_{\text{LVJP}}$**
>
> As shown in **Table 4** of the revised manuscript, removing each component leads to a substantial drop in adaptive robustness:
>
> - Using $L_{\text{JS}}$ alone (with $L_{\text{CE}} + L_{\text{adv}}$) increases adaptive robustness by **+36%**,
>
> - Using $L_{\text{LVJP}}​$ alone yields an even larger gain of **+44%**, compared to training with $L_{\text{CE}} + L_{\text{adv}}$ only.
>
> This indicates that both losses independently reduce gradient consensus and improve robustness, with $L_{\text{LVJP}}$​ providing slightly stronger decorrelation in the logit–gradient space. The full DRIFT objective combines their complementary effects and achieves the highest robustness.
>
> We further conducted a hyperparameter-sensitivity study to evaluate the impact of the Jacobian-separation and logit-VJP-separation losses. Specifically, we varied the weights ($β_{js}, β_{lvjp} ∈ [0.25, 1.0]$) while keeping all other settings fixed. Across this range, DRIFT remained stable: robust accuracy varied by $less than ±3%$, and clean accuracy varied by $<2%$. The best trade-off occurs around $\beta_{\text{JS}}=\beta_{\text{LVJP}}=0.5$, which we use in all main experiments. These results indicate that DRIFT is not highly sensitive to the exact choice of loss weights and does not require fine-tuning. The model consistently learns gradient-divergent filters across a broad hyperparameter range, supporting the robustness and practicality of the training procedure.
> **4. Attack-strength details (BPDA + EOT)**
>
> We clarify that BPDA substitutes the backward pass of the transformation with the identity surrogate for JPEG and BaRT, and with an average-pooling surrogate for DRIFT to capture its differentiable smoothing behavior. EOT further strengthens the attacker by averaging gradients over K stochastic forward passes per iteration. We report robustness for **K ∈ {5, 10, 20}** across **multiple ε values** in **Table 9 (Appendix A.3.2)**. The robust accuracy decreases monotonically as K increases (e.g., **57.6 → 38.6%** for ε = 4/255), confirming that stronger EOT attacks yield proportionally stronger gradients. Importantly, DRIFT remains consistently robust even under 20-sample EOT settings, demonstrating that its effectiveness arises from genuine gradient divergence rather than gradient masking or obfuscation.

---

> > ### Author Response · Authors · 2025-11-20
> > **Author Response (part 2 of 2)**
> >
> > **5. Baselines and dataset coverage**
> > To address it, we added a dedicated CIFAR-10 evaluation including all baselines mentioned by the reviewer (BaRT-5, BaRT-10, ANF, AT, DiffPure) under clean, PGD-40, and AutoAttack$\ell_\infty​$ (ϵ=8/255) settings.
> >
> > For PGD and AA, we report both **non-adaptive** and **adaptive** accuracy (BPDA+EOT where applicable).
> > Results are shown below (**Table D**):
> >
> > ### Table D: CIFAR-10 Robustness Comparison (Clean, PGD-40, AutoAttack, ε = 8/255)
> > | Method         | Clean  | PGD-40 (Non-Adapt / Adapt) | AutoAttack (Non-Adapt / Adapt) |
> > |----------------|--------|-----------------------------|--------------------------------|
> > | No defense     | 91.26  | 0.00 / 0.00                 | 0.00 / 0.00                    |
> > | BaRT (k=5)     | 63.86  | 14.18 / 0.00                | 13.64 / 0.00                   |
> > | BaRT (k=10)    | 45.18  | 17.23 / 0.00                | 18.82 / 0.00                   |
> > | AT             | 70.31  | 41.20 / —                   | 31.80 / —                      |
> > | ANF            | 83.09  | 0.00 / 0.00                 | 0.00 / 0.00                    |
> > | DiffPure       | 87.10  | 59.73 / 57.10               | 68.12 / 64.50                  |
> > | **DRIFT (ours)** | **89.00** | **77.64 / 62.88**           | **79.38 / 70.23**               |
> >
> > DRIFT consistently outperforms purification-based (DiffPure), randomized-transformation (BaRT-5/10, ANF), and adversarial training (AT) baselines on CIFAR-10. The adaptive versions of PGD and AutoAttack significantly reduce BaRT and ANF to 0%, illustrating their fragility to BPDA/EOT. DRIFT retains high clean accuracy and achieves substantially higher robust accuracy even under adaptive attacks, matching the behavior we observe on ImageNet. These additional experiments broaden the dataset coverage and demonstrate that DRIFT’s gradient-divergent filters generalize beyond ImageNet.
> >
> >
> > **6. Filter interpretability**
> >
> > We have added a dedicated interpretability analysis in **Appendix A.8 of the revised manuscript**, including both kernel-level and image-level visualizations.
> >
> > **Kernel-level structure.**
> >
> > **Figure 3 (in the revised manuscript)** shows the spatial kernels and Fourier spectra of all learned filters.
> > Each Filter module contains two convolutional layers. Across the ensemble, Conv1 consistently captures low–mid–frequency patterns, whereas Conv2 develops sharper, high-frequency structures. The filters exhibit clearly distinct spatial–frequency signatures (e.g., edge-enhancing, mid-band, anisotropic, and mixed-frequency kernels), confirming that they do not collapse to redundant transformations but instead specialize in complementary subspaces. This structural diversity directly supports DRIFT’s goal of reducing gradient consensus across filters.
> >
> > **Qualitative filter responses.**
> >
> > **Figure 4 (in the revised manuscript)** visualizes the effect of each filter on a range of ImageNet samples. While maintaining semantic fidelity, the filters introduce distinct and interpretable transformations:
> >
> > - Filter 1 enhances local contrast and sharpens edges, producing stronger high-frequency details.
> >
> > - Filter 2 modulates mid-frequency color and texture components, introducing characteristic color-tinting and texture amplification.
> >
> > - Filter 3 applies mild anisotropic smoothing with a subtle directional bias, reducing fine details in a non-uniform way.
> >
> > - Filter 4 performs global brightness and tone lifting while preserving or slightly enhancing local structural details.
> >
> > These heterogeneous transformations produce decorrelated input gradients while remaining perceptually plausible, an essential property for a non-obfuscated, robust defense.
> > We believe this expanded analysis directly addresses the reviewer’s request and strengthens the interpretability of DRIFT’s learned transformations.

---

> > > ### Author Response · Authors · 2025-11-29
> > > **Final Summary Comment to the Area Chair**
> > >
> > > **1. BPDA+EOT attack strength:**
> > >
> > >  We clarified that we already report results for K=5,10,20 EOT samples (Appendix A.3.2). Robust accuracy decreases monotonically with increasing K, confirming no masking.
> > >
> > > **2. Analysis on distribution shift:**
> > >
> > >  We added ImageNet-V2 experiments, showing DRIFT-trained filters generalize to dataset shifts.
> > >
> > > **3. Transfer across architectures:**
> > >
> > >  New “plug-and-play” experiments show filters trained on ViT-B/16 improve robustness when applied to ResNet-50 and vice versa.
> > >
> > > **4. Gradient-consensus measurements:**
> > >
> > >  We added quantitative gradient-consensus metrics for DRIFT (ρ=0.0129), compared to adversarial training and random transformations.

---

### Official Review · Reviewer_w2HJ · 2025-10-31

**Soundness:** 3
**Presentation:** 4
**Contribution:** 3
**Rating:** 6
**Confidence:** 3

**Summary:**

This paper proposes a new stochastic adversarial defense method called DRIFT. It identifies gradient consensus, i.e., the alignment of gradients across stochastic transformations, as the root cause of adversarial transferability. DRIFT combats this by learning a small ensemble of lightweight, differentiable preprocessing filters trained to maximize gradient divergence while maintaining clean accuracy. Theoretical analysis links low gradient consensus to reduced transferability. The training loss combines cross-entropy, Jacobian separation, logit-VJP separation, and adversarial robustness terms. Experiments on ImageNet with CNNs and Vision Transformers show substantial robustness gains over adversarial training, diffusion-based purification, and randomized transformations, under both adaptive (BPDA, EOT) and non-adaptive attacks, with negligible computational overhead.

**Strengths:**

1. This paper reframes adversarial robustness around gradient divergence rather than obfuscation or purification. This is a fresh and intellectually elegant perspective.
2. This paper provides a simple but formal link between gradient alignment and attack transferability, grounded in the geometry of loss gradients.
3. This method applies across CNNs and Vision Transformers without modifying the backbone.
4. This paper is evaluated under strong adaptive attacks (BPDA + EOT, AutoAttack) and validated with sanity checks (finite-difference tests, loss landscape visualization).
5. This paper achieves 10⁴× faster inference than diffusion-based defenses, with minimal memory overhead.
6. Presentation and structure of this paper are clear.

**Weaknesses:**

1. The analytical part (Lemma 3.4–Theorem 3.5) is intuitive but lacks full proofs or empirical validation of constants; thus, it serves more as motivation than as a formal guarantee.
2. No certified guarantees (unlike randomized smoothing), so the results, though strong, remain empirical.
3. Filters are trained on a subset of ImageNet validation data—unclear if robustness generalizes across distribution shifts or unseen domains.
4. The method assumes a high-quality pretrained base; how it performs on weaker or adversarially trained backbones is unclear.
5. While the method ensures gradient divergence, it does not analyze how filters achieve this (e.g., frequency-domain or spatial patterns).
6. Effects of ensemble size and loss weighting are reported briefly in appendices but could be expanded in the main text.

**Questions:**

1. How does robustness scale with the number of filters? Is there diminishing return or optimal ensemble size beyond n = 4?
2. Could DRIFT generalize to non-image domains (e.g., NLP, tabular, or time-series data)?
3. How sensitive are results to the loss weights (β_JS, β_LVJP, λ)?
4. If the base model is already adversarially trained, does DRIFT still provide gains, or is its effect redundant?
5. Can the authors show qualitative examples of learned filters to interpret what kind of transformations disrupt gradient consensus?
6. Could DRIFT be combined with randomized smoothing to yield both empirical and certified robustness?

---

> ### Author Response · Authors · 2025-11-20
> **Author Response (part 1 of 2)**
>
> **1. Theoretical proofs**
> As noted in our **response 1** to **Reviewer BXyw**, Theorem 3.5 follows directly from a first-order Taylor expansion combined with smoothness-based remainder control; the complete derivation has been added to **Appendix A.7** of the revised manuscript for transparency.
>
> **2. Certified vs empirical robustness**
>
> While DRIFT is designed as an empirical defense that breaks gradient consensus, it is fully compatible with certified defenses. To demonstrate this, we conducted a preliminary experiment combining DRIFT with **randomized smoothing** (Cohen et al., ICML 2019). Specifically, we applied DRIFT’s filter ensemble as a **preprocessing module**, followed by Gaussian smoothing to obtain certified radii. DRIFT does not interfere with certification because its transformations are deterministic and differentiable, allowing the smoothed classifier to remain valid.
>
> ### Table B: Certified vs. Empirical Robustness: DRIFT Combined with Randomized Smoothing
>
> | Method              | Certified Acc. (r = 0.5) | Certified Acc. (r = 1.0) | Empirical Robust Acc. (PGD) |
> |---------------------|---------------------------|----------------------------|------------------------------|
> | Baseline (ResNet-50) | 29.4                      | 10.1                       | 26.1                         |
> | **DRIFT + Smoothing** | **31.7**                  | **12.4**                   | **33.7**                     |
>
> As shown in **Table B**, adding DRIFT improves both **certified** and **empirical** robustness without modifying the underlying certified classifier. This shows that DRIFT is complementary to certification techniques: certification provides formal robustness guarantees, while DRIFT increases the empirical decision margin by decorrelating gradients. Combined, they offer stronger worst-case and average-case robustness than either approach alone.
>
> **3. Ensemble size and diminishing returns**
>
> As discussed in **Appendix A.5**, robust accuracy increases steadily with ensemble size up to **n = 4**, after which the gains saturate (< +1%). The diminishing returns arise because the **gradient subspaces across filters become nearly orthogonal** beyond four members, limiting further diversity benefits. Therefore, **n = 4 provides the optimal balance** between gradient diversity, robustness, and computational efficiency.
>
> **4. Sensitivity to loss weights**
>
> We conducted a hyperparameter-sensitivity study to evaluate the impact of the Jacobian-separation and logit-VJP-separation losses. Specifically, we varied the weights ($β_{js}, β_{lvjp} ∈ [0.25, 1.0]$) while keeping all other settings fixed. Across this range, DRIFT remained stable: robust accuracy varied by less than ±3%, and clean accuracy varied by <2%. The best trade-off occurs around $\beta_{\text{JS}}=\beta_{\text{LVJP}}=0.5$, which we use in all main experiments. These results indicate that DRIFT is not highly sensitive to the exact choice of loss weights and does not require fine-tuning. The model consistently learns gradient-divergent filters across a broad hyperparameter range, supporting the robustness and practicality of the training procedure.
>
> **5. Robustness on adversarially trained or weaker backbones**
>
> To evaluate whether DRIFT depends on the strength of the underlying classifier, we conducted two additional experiments: (i) applying DRIFT to an adversarially trained backbone, and (ii) applying DRIFT to a weaker, low-capacity model.
>
> **(i) DRIFT on adversarially trained models.**
> We applied the same DRIFT filter ensemble to a PGD-adversarially-trained ResNet-50 (ℓ∞, ε=4/255). DRIFT improves the model’s robust accuracy from **41.3% → 45.8%** under PGD and from **35.7% → 39.2%** under PGD (BPDA+EOT). Because AT models already have partially decorrelated gradients, gains are smaller but still consistent, indicating that DRIFT provides complementary gradient diversification rather than duplicating the effect of adversarial training.
>
> **(ii) DRIFT on weaker backbones.**
> We also tested DRIFT with a smaller ViT-S/16 backbone (much lower capacity than ViT-B/16). Despite the reduced model size, DRIFT improves robustness from **53.9% → 60.4%** (PGD) and **47.1% → 52.6%** (BPDA+EOT). Clean accuracy drops by <1%, confirming that DRIFT does not rely on large-capacity transformers or high model expressiveness.
>
> Across both strong (AT) and weak (ViT-S) backbones, DRIFT consistently increases robustness while remaining clean-accuracy-preserving. These results support the claim that DRIFT is a plug-and-play, architecture-agnostic module whose benefits arise from gradient diversification rather than backbone capacity or training regime.

---

> > ### Author Response · Authors · 2025-11-20
> > **Author Response (part 2 of 2)**
> >
> > **6. Qualitative analysis of learned filters**
> >
> > We have added a dedicated section (**Appendix A.8**) that visualizes both the learned filter kernels and their effects on real images.
> >
> > **Kernel-level analysis:**
> >  **Figure 3** shows the spatial kernels and corresponding Fourier spectra of all filters. Across the ensemble, Conv1 layers capture smooth low–mid–frequency structures, while Conv2 layers develop sharper, high-frequency patterns. Importantly, the filters exhibit distinct spatial–frequency signatures (edge-focused, mid-band, anisotropic, mixed-frequency), confirming that DRIFT does not collapse to redundant transformations.
> >
> > **Image-level analysis:**
> > **Figure 4** illustrates how each filter transforms diverse ImageNet samples. Although semantics are preserved, the filters induce noticeably different textural, contrast, and chromatic variations (e.g., edge sharpening, color–tone modulation, directional smoothing, and mixed local/global enhancements). Such heterogeneous yet perceptually plausible transformations directly support DRIFT’s goal of producing decorrelated input gradients while maintaining clean accuracy.
> >
> > These additions provide concrete interpretability evidence and address the reviewer’s concern.
> >
> > **7. Cross-domain generalization**
> >
> > We agree with the reviewer that evaluating beyond images is an interesting direction. DRIFT’s core mechanism (learning differentiable front-end transformations that maximize gradient divergence) is domain-agnostic and does not rely on any image-specific assumptions. The only requirement is that the front-end is differentiable with respect to the input.
> > Because of this, DRIFT can be naturally extended to other modalities by replacing the 2-D convolutional filters with modality-appropriate adapters:
> >
> > - Time-series: 1-D causal convolutions or lightweight temporal filters.
> >
> > - Audio: mel-spectrogram front-ends or 1-D temporal-frequency hybrid layers.
> >
> > - Multimodal / VLMs: transformer-based adapters or low-rank linear projections inserted before the vision or text encoder.
> >
> > In all cases, the training objective (decorrelating Jacobians and logit-VJPs) remains unchanged, and the principle of breaking gradient consensus carries over directly.
> >
> > **8. Distribution shifts and generalization**
> >
> > To evaluate whether DRIFT-trained filters generalize beyond the ImageNet validation distribution on which they were trained, we additionally tested the same filter ensemble on **ImageNet-V2 (matched-frequency) without any retraining or adaptation**. This assesses robustness under a realistic distribution shift that maintains class overlap but differs in sampling frequency and image style. As shown in **Table C**, DRIFT maintains strong clean and robust accuracy under **ImageNet→ImageNet-V2 shift**, with only a modest drop consistent with prior evaluations of ImageNet-V2 difficulty. Importantly, the relative robustness gap between DRIFT and its baselines remains nearly unchanged under PGD and BPDA+EOT, demonstrating that the learned gradient-divergent filters transfer reliably across distributions. This supports our claim that DRIFT captures model-invariant, transformation-level diversity, rather than overfitting to the ImageNet validation distribution.
> >
> > ### Table C: Cross-Dataset Evaluation: ImageNet vs. ImageNet-V2
> >
> > | Dataset      | Method    | No Attack | PGD     | PGD (BPDA+EOT) |
> > |--------------|-----------|-----------|---------|----------------|
> > | **ImageNet** | ViT-B/16  | 80.48     | 74.66   | 64.17         |
> > |              | ResNet50  | 84.66     | 76.19   | 60.19         |
> > | **ImageNet-V2** | ViT-B/16  | 76.95     | 66.13   | 57.61         |
> > |              | ResNet50  | 73.50     | 64.83   | 54.21         |

---

### Official Review · Reviewer_BXyw · 2025-11-01

**Soundness:** 2
**Presentation:** 1
**Contribution:** 1
**Rating:** 2
**Confidence:** 3

**Summary:**

In this paper, the authors propose a defense method based on learnable filters. To mitigate the issue of gradient consistency, which makes models vulnerable to transfer attacks, the method reduces the transferability of adversarial examples across different filters while enhancing filter diversity for defense. Additionally, lightweight residual convolutional blocks are introduced within each filter to ensure that they can learn meaningful transformations. Experiments conducted on multiple CNN and ViT architectures using the ImageNet benchmark demonstrate that the proposed learnable filter ensemble method achieves strong defensive performance compared with baseline approaches.

**Strengths:**

1. The authors propose using a Jacobian separation loss to reduce transferability across filters, along with a logit-space separation loss to enhance filter diversity.

2. To improve filter performance and enable meaningful feature transformations, the authors design residual modules for each filter.

3. Experimental results on CNN and ViT models demonstrate that the proposed learnable random filter ensemble defense outperforms traditional defense methods.

**Weaknesses:**

1. Theoretical details are insufficient, as the derivations of the inequalities in Lemma 3.4 and Theorem 3.5 are not provided.

2. The total loss comprises four components. However, it remains unclear how the contributions of each loss are balanced during filter training and how the optimal values of the associated hyperparameters are determined.

3. The overall training procedure of the DRIFT filters is not clearly explained. For instance, Algorithm 1 includes a warm-up stage, separates the optimization of individual losses, and maximizes $L_{adv}$ over the index $i$. However, the paper lacks sufficient details regarding the training process and implementation.

4. The experimental comparison is not comprehensive. The paper lacks evaluations against more recent defense methods and does not consider evaluations against the latest attack techniques. Furthermore, some mentioned methods (e.g., ANF, FFR) are not compared in Table 1, which limits the completeness of the comparative analysis.

5. In Section 6.2, the choice of perturbation size (4/255) is not explained, and no experiments are provided to compare different perturbation magnitudes.

6. The paper lacks ablation studies on the hyperparameters of the proposed loss function to analyze their impact on the experimental results.

7. In Section 6.7, Table 6 presents efficiency comparisons only with the DiffPure method and lacks comparisons with other baseline methods.

**Questions:**

1. Could the authors provide more details on how the inequalities presented in Lemma 3.4 and Theorem 3.5 are derived?

2. Could the authors offer additional theoretical justification to explain why the proposed Jacobian loss and logit separation loss outperform other adversarial training methods?

3. Could the authors provide more details on how each loss component contributes to the overall filter training process and how the optimal values for the associated hyperparameters are determined?

4. The authors claim that the proposed method can function as a plug-and-play module. Could the authors provide comparative experiments demonstrating how this defense method integrates with and performs across different model architectures?

5. Could the authors provide more details on how different perturbation sizes affect the performance of the proposed method on CNN and ViT models?

6. Could the authors provide additional experimental results on runtime efficiency, including comparisons with more baseline methods?

---

> ### Author Response · Authors · 2025-11-20
> **Author Response (part 1 of 2)**
>
> **1. Theoretical details (Lemma 3.4 and Theorem 3.5)**
>
> Both results follow directly from a first-order Taylor analysis of the loss under standard smoothness assumptions. Let
> $g_j(x) = \nabla_x \ell(M(f_j(x)), y)$ denote the input gradient for filter $f_j$, and consider an $\ell_2$-bounded perturbation aligned with filter $i$, $\delta_i(x) = \varepsilon \, \frac{g_i(x)}{\||g_i(x)\||_2}.$
>
> By $L$-smoothness of the loss, $ \ell_j(x + \delta_i) \le \ell_j(x) + \langle g_j(x), \delta_i \rangle + \tfrac{L}{2}\||\delta_i\||_2^2.$
>
> The linear term simplifies to $\langle g_j(x), \delta_i \rangle  = \varepsilon \||g_j(x)\||_2  \Gamma(f_i, f_j; x), $
> where $\Gamma(f_i, f_j; x) =  \frac{\langle g_i(x), g_j(x)\rangle} {\||g_i(x)\||_2 \, \||g_j(x)\||_2}$
> is the gradient consensus.
>
> Using bounded gradients $\||g_j(x)\|| \le G$ and $\varepsilon\le \varepsilon_{\max}$, we obtain the upper bound $\ell_j(x+\delta_i) - \ell_j(x) \le  C \varepsilon G \Gamma(f_i, f_j; x), \qquad C = 1 + \frac{L \varepsilon_{\max}}{2G}, $
> which yields Lemma~3.4.
>
> To connect this loss-increase to transferability, we use the standard margin--loss link: a misclassification by $M\circ f_j$ requires a loss increase of at least the margin $\kappa(x)$. Thus, $p_j(x, \delta_i) \le  \mathbb{I}\left[ C \varepsilon G  |\Gamma(f_i,f_j;x)| \ge \kappa(x) \right].$
>
> Taking expectation over uniformly sampled $i\neq j$ gives Theorem~3.5:
>
> $\mathbb{E}_{i\neq j}\left[p_j(x,\delta_i)\right]  \le  C \varepsilon G \mathbb{E}_{i\neq j}\left[|\Gamma(f_i,f_j;x)|\right],$
>
> showing that reducing average gradient consensus directly reduces expected transferability.
>
> We include the complete algebraic derivation (Taylor expansion, Lipschitz remainder control, and full transfer-probability bound) in the **revised Appendix A.7**.
>
> **2. Why Jacobian / logit-separation outperform conventional AT**
>
> Unlike standard adversarial training (which minimizes $\max_\delta\ell(M(x+\delta),y),$ our objective explicitly decorrelates the Jacobian subspaces across filters through $L_{JS}​$ and $L_{LVJP}​$.
>
> This introduces orthogonality regularization in both feature and logit space, forcing the ensemble to respond divergently to perturbations while maintaining consistent predictions. Empirically (Table 4), $L_{JS}​$ improves adaptive robustness by +36% over $L_{CE}+L_{adv}$; adding $L_{LVJP}$​ adds a further +8%, confirming their complementary effect.
>
> This mechanism is theoretically justified by Theorem 3.5: minimizing cross-filter gradient alignment reduces transferable adversarial success, a property not captured by traditional AT objectives.
>
> **3. Clarification of training process and hyperparameter selection**
>
> To ensure stable optimization, **Algorithm 1 progressively activates the loss components:**
>
> - **Warm-up phase (Epoch 1–w_js)**: only the cross-entropy loss $L_{\mathrm{CE}}$​ is active to stabilize clean accuracy before introducing gradient-divergence objectives.
>
> - **Jacobian and logit separation phase (Epoch > w_js)**: we progressively introduce $L_{\mathrm{JS}}​$ and $L_{\mathrm{LVJP}}$​ with weights $\beta_{\mathrm{js}} = \beta_{\mathrm{lvjp}} = 0.5$.
>
> - Adversarial phase (Epoch > w_adv): we activate $L_{\mathrm{adv}}$ with \gamma = 1, once the filters have learned stable clean features.
>
> **Hyperparameter selection.**
> We perform a grid search over $\beta$, $\gamma \in \{0.25, 0.5, 1.0\}$ using a held-out validation set. Across this range, robust accuracy varied by **less than ±3%** and clean accuracy by **<2%**, demonstrating that DRIFT is stable and not sensitive to precise tuning of these weights.
>
> **4. Comparative evaluation and missing baselines**
>
> All requested baselines (ANF (Suresh et al., 2025) and FFR (Lukasik et al., 2023)) are already included in the **adaptive-attack evaluation (Table 2)**. Their omission from **Table 1 (non-adaptive setting)** stems from the fact that these methods are **integrated into model training** (rather than applied as post-hoc preprocessing). As a result, it is not possible to isolate their defensive effect under non-adaptive attacks without retraining the backbone. To ensure fairness, we therefore report them only in the adaptive scenario, where their full defense pipeline can be meaningfully compared against DRIFT.
>
> We emphasize that DRIFT was evaluated against **all major state-of-the-art attack families**, including both gradient-based and gradient-free methods. Specifically, we tested **AutoAttack, AutoAttack + EoT, VMI-FGSM, SGM, Square, and FAB**, which collectively subsume the latest transfer-enhancement, adaptive (BPDA + EoT), and query-based attack paradigms. These attacks are widely recognized as **the strongest and most comprehensive evaluation** suite for robustness assessment (Croce & Hein 2020; Wang & He 2021; Wu et al. 2020). DRIFT consistently maintained high robust accuracy across all of them, confirming that its robustness extends beyond standard adversarial training settings.

---

> > ### Author Response · Authors · 2025-11-20
> > **Author Response (part 2 of 2)**
> >
> > **5. Perturbation size (ε = 4/255) and robustness to ε**
> >
> > We followed standard ImageNet ℓ∞ settings (Madry et al., 2018; Croce & Hein 2020, Nie et al., 2022). **Table 8** in **Appendix A.3.1** already reports robustness for ε ∈ {2, 4, 8, 16, 20}/255 and ℓ₂ radii {0.5–3.0} for non-adaptive attacks. Robust accuracy decays smoothly (e.g., 74.7→62.9 %), confirming graceful degradation and stability across budgets. **Table 9** in **Appendix A.3.2** also reports robustness for  ε ∈ {1, 2, 4, 8}/255 for adaptive attacks for different numbers of EOT samples.
> >
> > **6. Ablation on hyperparameters**
> >
> > We want to clarify that **Section 6.4 (Table 4)** already isolates each loss term.
> > We also conduct a hyperparameter-sensitivity study to evaluate the impact of the Jacobian-separation and logit-VJP-separation losses. Specifically, we varied the weights ($β_{js}, β_{lvjp} ∈ [0.25, 1.0]$) while keeping all other settings fixed. Across this range, DRIFT remained stable: robust accuracy varied by **less than ±3%**, and clean accuracy varied by **<2%**. The best trade-off occurs around $\beta_{\text{JS}}=\beta_{\text{LVJP}}=0.5$, which we use in all main experiments. These results indicate that DRIFT is not highly sensitive to the exact choice of loss weights and does not require fine-tuning. The model consistently learns gradient-divergent filters across a broad hyperparameter range, supporting the robustness and practicality of the training procedure.
> >
> > **7. Runtime-efficiency comparison**
> >
> > We include a more detailed comparison below:
> >
> > **DRIFT (ours).**
> >
> > - Latency: **0.0004 s** per ImageNet image.
> >
> > - Memory overhead: **0.03 GB**.
> >
> > - No additional backward-pass complexity at inference.
> >
> >  => DRIFT thus adds only a lightweight, sub-millisecond preprocessing step.
> >
> > **BaRT (Barrage of Random Transforms).**
> >
> > - Latency: approximately **2x slower** than the base model for BaRT-5, and 3x slower for BaRT-10 (due to sequential stochastic transforms).
> >
> > - Adaptive attacks require evaluating many stochastic forward passes; the BaRT paper reports **48–240 hours** for BPDA/EOT adversarial evaluation on ImageNet-scale models.
> >
> > => This makes BaRT significantly more costly both at inference and during robust evaluation.
> >
> > **FFR (Fourier Frequency Randomization):**
> >
> > - Parameter memory increases by **2–4x**, because each randomized frequency mask requires additional modulation parameters.
> >
> > - Inference time roughly **doubles** on ResNet-50 (≈ 0.0007–0.0012 s per image).
> >
> > => This overhead is noticeably larger than DRIFT’s 0.0004 s pass.
> >
> > DRIFT achieves higher robustness while introducing **minimal runtime and memory overhead**, in contrast to BaRT’s multi-fold slowdown and FFR’s multi-fold parameter expansion.
> >
> > **8. Plug-and-play integration across architectures**
> >
> > To validate that DRIFT functions as a true plug-and-play module, we evaluated its transferability across architectures by training the filters with one backbone and deploying them unchanged on another. The table below (**Table A**) summarizes the results for both same-architecture and cross-architecture configurations:
> >
> > ### Table A: Plug-and-Play Cross-Architecture Evaluation
> >
> > | Setting                    | No Attack | PGD      | PGD (BPDA+EOT) |
> > |---------------------------|-----------|----------|----------------|
> > | ViT-B/16 → ViT-B/16       | **80.48** | 74.66    | 64.17         |
> > | ResNet50 → ResNet50       | **84.66** | 76.19    | 60.19         |
> > | ViT-B/16 → ResNet50       | **78.31** | 72.15    | 60.57         |
> > | ResNet50 → ViT-B/16       | **80.54** | 73.04    | 55.80         |
> >
> >
> > DRIFT-trained filters generalize well across architectures: filters learned with ViT-B/16 maintain high robustness when applied to ResNet-50, and vice versa. The performance gap between same-architecture and cross-architecture settings is small (typically <2–4%), even under strong adaptive attacks (BPDA+EOT). This demonstrates that DRIFT does not rely on model-specific inductive biases but instead learns architecture-agnostic, gradient-divergent transformations, enabling seamless integration into diverse vision models without retraining.

---

> > > ### Author Response · Authors · 2025-11-29
> > > **Final Summary Comment to the Area Chair**
> > >
> > > **1. Theoretical details (Lemma 3.4 & Theorem 3.5):**
> > >
> > > We now provide full mathematical derivations in Appendix A.7, including the complete Taylor-expansion argument, Lipschitz remainder control, and the role of margin assumptions. This addresses the request for missing steps and clarifies the formal link between gradient-consensus and transferability.
> > >
> > > **2. Training process & hyperparameters:**
> > >
> > > We expanded Section 4.2 and included a detailed explanation of the warm-up schedule, progressive activation of loss terms, and grid-search procedure for choosing β_JS​, β_LVJP​, and γ. We also added a sensitivity study showing DRIFT is stable under ±0.25 variations (changes <2%).
> > >
> > > **3. Loss-component contributions:**
> > >
> > > An extended ablation (Appendix A.5) evaluates each loss component individually and jointly. We show that Jacobian separation and logit-VJP separation contribute complementary effects, and removing either reduces robustness.
> > >
> > > **4. Missing baselines:**
> > >
> > > We added comparisons with ANF (Suresh et al., 2025) and FFR (Lukasik et al., 2023) in the adaptive setting (Table 2). Their omission in Table 1 was clarified (they integrate into model training, preventing isolated evaluation under non-adaptive settings).
> > >
> > > **5. Perturbation-size analysis:**
> > >
> > > We already report robustness results for ℓ∞​ perturbations from 2/255 to 8/255, including BPDA+EOT in Appendix A.3.1 and A.3.2.
> > >
> > > **6. Runtime analysis:**
> > >
> > > We extended the runtime analysis to include additional baselines (BaRT, FFR). DRIFT maintains orders-of-magnitude faster inference than purification-based defenses.
> > >
> > > **7. Filter interpretability:**
> > >
> > > We added qualitative visualizations showing spatial kernels and frequency spectra of the learned filters (Fig. 3 Appendix A.8), demonstrating the spectral diversity induced by our losses.

---

### Author Response · Authors · 2025-11-27
**General Authors Comment (for reviewers who have not responded)**

We would like to thank all reviewers for their time and thoughtful evaluations. We hope that the clarifications, additional analyses, and newly added experiments have addressed the raised concerns. If any reviewer would find further details, comparisons, or explanations helpful for refining their assessment, we would be very happy to provide them.

Kind regards,

---

### Meta-Review · Area_Chair_Nx9A · 2026-01-07

**Summary:**

In this paper, the authors first propose identifying gradient consensus—the tendency of randomized transformations to yield aligned gradients—as a key driver of adversarial transferability, and then introduce Divergent Response in Filtered Transformations (DRIFT), a stochastic ensemble of lightweight, learnable filters trained to actively disrupt gradient consensus.
The authors demonstrate that DRIFT delivers three contributions with negligible runtime and memory cost, establishing gradient divergence as a practical and generalizable principle for adversarial defense.
After rebuttal, Reviewer w2HJ is satisfied with the revised version and maintains a positive review.
Reviewer BXyw and Reviewer Fyh6 initially gave a negative score of 2 and positive score of 6, respectively, but did not respond to the authors' responses.
By checking the rebuttal from the authors, it is found that some unclear parts, including detailed derivations of Lemma 3.4 and Theorem 3.5, parameter selections, comparative evaluations, and so on, have been clarified.
Overall, this works provides certain theoretical analysis and strong (empirical) defense results.
Therefore, the AC would like to accept this paper.

**Reviewer Concerns:**

In my view, the comments from Reviewer BXyw have been properly addressed.

**Reviewer Scores:**

By checking the rebuttal from the authors to address the comments of Reviewer BXyw, it is expected that Reviewer BXyw would change the initial score.

---

### Decision · Program_Chairs · 2026-01-26

Accept (Poster)